# Bayesian Test-Time Adaptation via Dirichlet feature projection and GMM-Driven Inference for Motor Imagery EEG Decoding

**Huan Luo[1], Na Lu[2],\* Xiaopeng Wang[2], Xu Niu[2]**
[1]School of Future Technology, Xi'an Jiaotong University
[2]School of Automation Science and Engineering, Xi'an Jiaotong University
`luohuan123@stu.xjtu.edu.cn, lvna2009@xjtu.edu.cn`

## Abstract

Generalization in EEG-based motor imagery (MI) brain-computer interfaces (BCIs) is hampered by cross-subject and cross-session variability. Although large-scale EEG pretraining has advanced representation learning, their practical deployment is hindered by the need for costly fine-tuning to overcome significant domain shifts. Test-time adaptation (TTA) methods that adapt models during inference offer a promising solution. However, existing EEG-TTA methods either rely on gradient-based fine-tuning (suffering from high computational cost and catastrophic forgetting) or data alignment strategies (failing to capture shifts in temporal predictive embeddings). To address these limitations, we propose BTTA-DG, a novel Bayesian Test-Time Adaptation framework that performs efficient, gradient-free adaptation by modeling the distribution of temporal predictive embeddings. Our approach first employs a lightweight SincAdaptNet with learnable filters to extract task-specific frequency bands. We then introduce a novel Dirichlet feature projection that maps temporal embeddings onto a compact and interpretable parameter space, effectively capturing the concentration of time-varying predictive evidence. Adaptation is achieved via a GMM-driven Bayesian inference mechanism, which models the historical distribution of these Dirichlet parameters and fuses this evidence with the model's prior predictions to calibrate outputs for the target domain. Extensive experiments show that BTTA-DG significantly outperforms previous EEG-TTA methods, achieving state-of-the-art accuracy while running at real-time speed. Furthermore, visualizations confirm the physiological interpretability of our learned filters and the robust class separability of our Dirichlet feature space.

## 1 Introduction

Electroencephalography (EEG)–based brain–computer interfaces (BCIs) decode cortical activity to control external devices (Clerc, 2013). Among them, motor-imagery (MI) BCIs leverage sensorimotor rhythms (SMRs) (Neuper et al., 2006) and have shown promise in stroke rehabilitation (López-Larraz et al., 2018) and assistive control (Fernández-Rodríguez et al., 2016; Noda et al., 2012). Recent advances in large-scale EEG pretrained models (Wang et al., 2024; Jiang et al., 2024; Kim et al., 2024), trained on massive datasets, have demonstrated unprecedented capabilities in learning general and reliable representations. However, deploying such models in real-world scenarios remains challenging due to significant data shifts in EEG recordings (Huang et al., 2023). These shifts stem primarily from the non-stationarity of EEG signals, driven by cross-subject or cross-session neurophysiological differences (Apicella et al., 2024). Consequently, bridging these distributional gaps requires robust adaptive strategies beyond pretraining alone (Xu et al., 2020; Wimpff et al., 2025; Liu et al., 2025).

Test-time adaptation (TTA), which adapts models during inference using online unlabeled data (Li et al., 2023), presents a promising solution for practical BCI deployment. However, existing EEG-TTA methods fall into two paradigms with critical trade-offs. Gradient-based approaches update model parameters through techniques such as entropy minimization (Wang et al., 2020), pseudo-

---

*Corresponding author.

label optimization (Lee et al., 2013; Wang et al., 2022), and consistency regularization. While effective, these methods suffer from high computational overhead due to gradient backpropagation and risk catastrophic forgetting when continuously updating pre-trained representations. For instance, OTTA (Wimpff et al., 2024) integrates data alignment with entropy-based batch normalization finetuning, while T-TIME (Li et al., 2023) employs ensemble learning with conditional entropy minimization, both requiring substantial computational resources. Conversely, non-gradient approaches avoid parameter updates by recalibrating domain-specific statistics, such as batch normalization layer recalculation (Schneider et al., 2020) or data alignment (Wimpff et al., 2024; Bakas et al., 2025). While computationally efficient, these methods rely on shallow alignment techniques that inadequately capture the complex, domain shifts inherent in EEG representations across subjects and sessions. This creates a fundamental challenge: to develop a TTA framework that is both computationally efficient and capable of modeling deep distributional changes, all while being theoretically grounded and avoiding destructive model updates.

To address these challenges, we propose BTTA-DG, a Bayesian Test-Time Adaptation framework that achieves high-performance, gradient-free adaptation via Dirichlet feature projection and GMM-driven inference. Our approach introduces the Dirichlet distribution to EEG-TTA, treating it as a "distribution over categorical distributions" to model prediction uncertainty (Wong, 1998). Our method first employs a Sinc-based adaptive network (SincAdaptNet) to extract powerful, task-specific features. We then introduce a novel Dirichlet feature projection, which maps temporal predictive embeddings onto a compact, interpretable parameter space. Unlike conventional EEG-TTA methods that rely on heuristic data (Zanini et al., 2017; He & Wu, 2019) or shallow statistics alignment (Schneider et al., 2020), this probabilistic representation effectively models the concentration of the model's time-varying predictive distribution, allowing us to capture shifts in temporal predictive embeddings in the new domain, which is a significant advance. To ensure stable and efficient adaptation, we design a Gaussian Mixture Model (GMM)-driven Bayesian inference mechanism. A GMM models the historical distribution of Dirichlet parameters from the target domain, and Bayesian inference fuses this likelihood with the deep model's prior predictions. This entire process is gradient-free, calibrating the model's outputs without destructive updates to its pre-trained weights, thereby preventing catastrophic forgetting. Our contributions include:

- We construct a Sinc-based adaptive network (SincAdaptNet) that leverages learnable Sinc-filters to extract task-specific frequency bands. It enhances the representation of temporal embeddings by isolating the most informative spectral components.

- We are the first to introduce the Dirichlet distribution to EEG-TTA, creating a low-dimensional projection that provides a robust and interpretable representation of deep distributional shifts, overcoming the limitations of prior heuristic and shallow alignment methods.

- We propose a novel GMM-driven Bayesian inference mechanism that enables gradient-free adaptation. By modeling the historical Dirichlet parameter distribution, the GMM retains global-neighborhood knowledge of the test data. Bayesian inference then combines the GMM likelihood with prior predictions to yield calibrated posterior predictions.

Across public MI datasets, BTTA-DG achieves state-of-the-art accuracy with real-time speed. Visualization analyzes confirm the physiological interpretability of our learned spatial and spectral filters, which isolate MI-specific scalp topographies (frontal, central, parietal, and occipital regions) and frequency bands (mu, beta, gamma). Furthermore, we show that our Dirichlet feature space yields robust class separability, evidenced by low intra-class covariance ($<0.27$) and high inter-class KL divergence ($>31.85$). These results validate BTTA-DG as a lightweight, robust, and theoretically grounded framework for practical test-time adaptation in BCIs.

## 2 METHODOLOGY

**Notations** In MI-TTA research, a cross-subject setting is commonly considered, where leave-one-subject-out (LOSO) cross-validation is widely adopted (Altaheri et al., 2023). Each subject in turn serves as the unlabeled target, with all other subjects' data forming the source. Let $\{D_{\text{src}}^l\}_{l=1}^{L}$ denote the source domain containing labeled EEG trials from $L$ subjects, where $D_{\text{src}}^l = \{(\boldsymbol{s}_l^i, \boldsymbol{y}_l^i)\}_{i=1}^{N^l}$ consists of $N^l$ trials for subject $l$. Each trial $\boldsymbol{s}_l^i \in \mathbb{R}^{C \times T}$ is a $C$-channel EEG signal of length $T$,

with class label $\boldsymbol{y}_l^i \in \mathcal{L}$. The target domain $D_{\mathrm{tgt}} = \{\boldsymbol{s}^i\}_{i=1}^{N_{\mathrm{tgt}}}$ contains $N_{\mathrm{tgt}}$ unlabeled test trials arriving sequentially from one single subject. A deep classification model $f_{\boldsymbol{\theta}} = f_{\mathrm{cls}} \circ g_{\mathrm{enc}}$, with parameters $\boldsymbol{\theta}$, is pre-trained on $\{D_{\mathrm{src}}^l\}_{l=1}^L$. Here, the encoder $g_{\mathrm{enc}} : \mathbb{R}^{C \times T} \to \mathbb{R}^{|\mathcal{L}| \times T}$ maps the EEG trials to embeddings of dimension $|\mathcal{L}| \times T$, and the classifier $f_{\mathrm{cls}} : \mathbb{R}^{|\mathcal{L}| \times T} \to \mathbb{R}^{|\mathcal{L}|}$ produces predictions for $|\mathcal{L}|$ classes. The goal is to adapt $f_{\boldsymbol{\theta}}$ to $D_{\mathrm{tgt}}$ via online MI-TTA framework without requiring target labels or source data. In addition, within-subject adaptation across sessions is also discussed in Appendix H.

## 2.1 Sinc-based Adaptive Bandpass Filtering Network

We propose a lightweight Sinc-based Adaptive Bandpass Filtering Network (SincAdaptNet) to serve as the deep classification model $f_{\boldsymbol{\theta}}$ within the MI-TTA framework. The network comprises: Spat-Conv $\to$ Sinc-Conv $\to$ IncCh-Conv $\to$ Cls-Conv, with layer normalization inserted after temporal filtering and channel expansion to avoid batch-statistics dependence when the online batch size is one (Ba et al., 2016).

**Spat-Conv** performs spatial filtering with $F_{\mathrm{spat}}$ kernels of size $(C \times 1)$, reducing channel redundancy while retaining task-relevant spatial patterns, akin to data-driven CSP (Blankertz et al., 2007). **Sinc-Conv** is an interpretable, parametrized temporal convolution inspired by SincNet (Ravanelli & Bengio, 2018; Zhang et al., 2024): rather than learning free-form kernels, it learns low cutoff $f_{\mathrm{low}}$ and bandwidth $f_{\mathrm{band}}$ (thus $f_{\mathrm{high}} = f_{\mathrm{low}} + f_{\mathrm{band}}$), from which a windowed-sinc band-pass kernel is generated. This yields MI-relevant mu (8–13 Hz), beta (13–30 Hz) McFarland et al. (2000); Pfurtscheller et al. (2006), and gamma ($> 30$ Hz) Darvas et al. (2010) rhythms with few parameters and clear spectral interpretability. The Sinc-Conv layer comprises $F_{\mathrm{sinc}}$ adaptive bandpass filters of size $(1 \times N_{\mathrm{sinc}})$ and uses "SAME" padding to preserve temporal dimension. **IncCh-Conv** expands channels to $2F_{\mathrm{sinc}}$ to enrich representation, and **Cls-Conv** maps features to $|\mathcal{L}| \times T$ embeddings, where $|\mathcal{L}|$ denotes the number of motor imagery classes and $T$ denotes the temporal length.

Critically, inspired by the SwAV framework (Caron et al., 2020), normalized probability spaces enable more stable and interpretable representation learning under domain shift. For each EEG trial, its temporal predictive embeddings are mapped through a softmax function to follow time-varying categorical distributions, denoted by $\boldsymbol{X} = g_{\mathrm{enc}}(\boldsymbol{s}) = [\boldsymbol{x}_1, \boldsymbol{x}_2, \ldots, \boldsymbol{x}_T] \in \mathbb{R}^{|\mathcal{L}| \times T}$. Each $\boldsymbol{x}_j \in \mathbb{R}^{|\mathcal{L}|}$ encodes an instantaneous class probability vector at timestep $j$, collectively forming a trajectory of time-varying uncertain predictions. The model's prior prediction is obtained through temporal averaging: $f_{\mathrm{cls}}(\boldsymbol{X}) = \frac{1}{T} \sum_{j=1}^T \boldsymbol{x}_j \in \mathbb{R}^{|\mathcal{L}|}$. This averaging process integrates the dynamic prediction information over time, providing a robust and informative prior for subsequent Bayesian inference. Complete architectural details of SincAdaptNet are provided in Appendix B.

## 2.2 Bayesian Test-time Adaptation via Dirichlet Feature Projection and GMM-driven Inference

To address challenges in current MI-TTA methods–catastrophic forgetting from gradient-based updates and insufficient modeling of deep feature shifts–we propose a probabilistic test-time adaptation framework that integrates Dirichlet feature projection with GMM-driven Bayesian inference. This approach dynamically calibrates the deep model's prior predictions $f_{\boldsymbol{\theta}}(\boldsymbol{s})$ through Bayesian inference, enabling efficient, gradient-free and theoretically grounded online test-time adaptation.

Overall, the Dirichlet feature projection method first projects the post-softmax temporal predictive embeddings $\boldsymbol{X}$ to the low-dimensional Dirichlet parameters. It preserves the prior concentration towards each class for temporal predictive embeddings that follow a time-varying categorical distribution. Second, in GMM-driven Bayesian inference, the Dirichlet parameters estimated from historical EEG trials are clustered by a GMM, which efficiently encodes both global information of the historical test trials and the neighborhood information of the current calibrating trial. Bayesian inference is then used to combine the GMM likelihood with the prior prediction of deep model to obtain the calibrated posterior.

## 1) Dirichlet Feature Projection for Deep-feature Modeling

The core of our method is the Dirichlet feature projection. The Dirichlet distribution, as a "distribution over distributions (Wong, 1998)", is a probability distribution over the parameter space

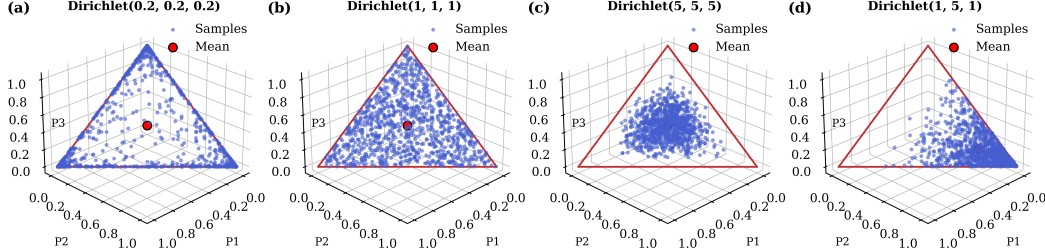

Figure 1: Dirichlet distributions for varying parameter settings. Each subplot displays 1000 samples (blue points) and the mean (red points) within the probability simplex (red lines). (a)-(c) illustrate that increasing the total scale $\alpha_0$ yields lower uncertainty (lower variance). Comparing (b) and (d) illustrates that elevating a component $\alpha_i$ shifts the prior concentration toward its corresponding class.

of categorical distribution. It can encode temporal predictive embeddings of categorical distributions for an EEG trial rather than a fixed categorical distribution. Let the temporal predictive embeddings output from SincAdaptNet's encoder be $\boldsymbol{X} = [\boldsymbol{x}_1, \boldsymbol{x}_2, \ldots, \boldsymbol{x}_T] \in \mathbb{R}^{|\mathcal{L}| \times T}$, where each $\boldsymbol{x}_j = (x_{1j}, x_{2j}, \ldots, x_{|\mathcal{L}|j})^\top \in \Delta^{|\mathcal{L}|-1}$ represents instantaneous categorical probability vector at timestep $j$. The probability simplex $\Delta^{|\mathcal{L}|-1}$ is defined as

$$\Delta^{|\mathcal{L}|-1} = \left\{ \boldsymbol{x}_j \in \mathbb{R}^{|\mathcal{L}|} : \sum_{i=1}^{|\mathcal{L}|} x_{ij} = 1, \ x_{ij} \geq 0 \right\}. \tag{1}$$

The Dirichlet distribution offers interpretable parameter $\boldsymbol{\alpha}$, where each component $\alpha_i$ reflects the concentrated prior probability towards class $i$ of temporal predictive embeddings, and the scale $\alpha_0 = \sum_{i=1}^{|\mathcal{L}|} \alpha_i$ indicates the overall uncertainty across $T$ time steps (Ng et al., 2011) (see Figure 1). It is important to note that $\boldsymbol{\alpha}$ (and thus $\alpha_0$) is obtained by maximum-likelihood estimation via a fixed-point algorithm rather than by manual tuning.

Assume that the temporal predictive embeddings $\boldsymbol{X}$ follow a Dirichlet distribution, denoted as $\boldsymbol{X} \sim \text{Dir}(\boldsymbol{\alpha})$, with parameter vector $\boldsymbol{\alpha} = (\alpha_1, \ldots, \alpha_{|\mathcal{L}|}) \in \mathbb{R}_+^{|\mathcal{L}|}$. Each categorical probability vector $\boldsymbol{x}_j$ is an i.i.d. sample from $\text{Dir}(\boldsymbol{\alpha})$. The support of $\text{Dir}(\boldsymbol{\alpha})$ is also confined to $\Delta^{|\mathcal{L}|-1}$.

We define a projection $\mathcal{P}$ that maps the temporal predictive embeddings $\boldsymbol{X} \in \mathbb{R}^{|\mathcal{L}| \times T}$ into its low-dimensional Dirichlet parameters $\boldsymbol{\alpha} \in \mathbb{R}_+^{|\mathcal{L}|}$ via maximum likelihood estimation (MLE),

$$\mathcal{P} : \mathbb{R}^{|\mathcal{L}| \times T} \to \mathbb{R}_+^{|\mathcal{L}|}, \quad \boldsymbol{X} \mapsto \hat{\boldsymbol{\alpha}}_{\text{MLE}} = \arg\max_{\boldsymbol{\alpha}} \sum_{j=1}^{T} \log \mathcal{D}(\boldsymbol{x}_j; \boldsymbol{\alpha}), \tag{2}$$

where the Dirichlet probability density function is given by

$$\mathcal{D}(\boldsymbol{x}_j; \boldsymbol{\alpha}) = \frac{\Gamma(\alpha_0)}{\prod_{i=1}^{|\mathcal{L}|} \Gamma(\alpha_i)} \prod_{i=1}^{|\mathcal{L}|} x_{ij}^{\alpha_i - 1}. \tag{3}$$

Here, $\Gamma(\cdot)$ is the Gamma function.

This projection effectively compresses the temporal dynamics of the deep features into a single, semantically rich vector that parameterizes the model's predictive distribution for that trial. The Dirichlet parameter estimate, $\hat{\boldsymbol{\alpha}}_{\text{MLE}}$, could be efficiently computed using a established fixed-point iteration algorithm (Minka, 2012), detailed in

$$\alpha_i^{\text{new}} = \psi^{-1} \left( \psi(\alpha_0^{\text{old}}) + \frac{1}{T} \sum_{j=1}^{T} \log x_{ij} \right), \tag{4}$$

where $\psi(u) = \frac{d}{du} \Gamma(u)$ denotes the Digamma function. Full algorithmic details for computing the post-projection Dirichlet parameters are provided in Appendix C.

2) GMM-DRIVEN BAYESIAN INFERENCE FOR GRADIENT-FREE CALIBRATION

For historical test EEG trials $\boldsymbol{s}$, we compute their Dirichlet parameters via the projection $\hat{\boldsymbol{\alpha}}_{\mathrm{MLE}} = \mathcal{P}(g_{\mathrm{enc}}(\boldsymbol{s}))$ and store parameters of high-confidence trials in a memory bank $M_y$ organized by their (calibrated) predicted label. This Dirichlet feature projection is performed independently for each trial and depends only on its own temporal predictive embeddings. A Gaussian Mixture Model (GMM) (Reynolds et al., 2009) is then employed to cluster the historical Dirichlet parameters in $M_y$ for each class to build a non-parametric density estimate of the historical Dirichlet parameters accumulated from the target domain, yielding a class-specific GMM likelihood:

$$p_{\mathrm{GMM}}(\boldsymbol{\alpha} \mid y) = \sum_{k=1}^{K} \pi_{y,k} \, \mathcal{N}(\boldsymbol{\alpha}; \boldsymbol{\mu}_{y,k}, \boldsymbol{\Sigma}_{y,k}), \tag{5}$$

where $K$ is the number of mixture components for class $y$, $\pi_{y,k}$ are the weights satisfying $\sum_{k=1}^{K} \pi_{y,k} = 1$ and $\pi_{y,k} > 0$, and $\boldsymbol{\mu}_{y,k}$ and $\boldsymbol{\Sigma}_{y,k}$ are the mean and covariance of the $k^{\mathrm{th}}$ Gaussian component, respectively.

The GMM encodes the global distribution of the historical test EEG trials and each component model preserves the neighborhood information of the current test trial to be calibrated. Any proximity of the current test trial to a particular cluster results in a relatively large GMM likelihood.

For a current test EEG trial $\boldsymbol{s}^i$ with parameter $\hat{\boldsymbol{\alpha}}_{\mathrm{MLE}}$, the calibrated posterior is computed by combining the GMM likelihood $p_{\mathrm{GMM}}(\hat{\boldsymbol{\alpha}}_{\mathrm{MLE}} \mid y)$ with the deep model's prior prediction $p_{\boldsymbol{\theta}}(y) = f_{\boldsymbol{\theta}}(\boldsymbol{s}^i)$ via Bayesian inference:

$$p_{\mathrm{cal}}(y \mid \hat{\boldsymbol{\alpha}}_{\mathrm{MLE}}) = \frac{p_{\mathrm{GMM}}(\hat{\boldsymbol{\alpha}}_{\mathrm{MLE}} \mid y) \, p_{\boldsymbol{\theta}}(y)}{\sum_{y'=1}^{|\mathcal{L}|} p_{\mathrm{GMM}}(\hat{\boldsymbol{\alpha}}_{\mathrm{MLE}} \mid y') \, p_{\boldsymbol{\theta}}(y')}. \tag{6}$$

Thus, the historical GMM influences only the posterior through the term $p_{\mathrm{GMM}}(\hat{\boldsymbol{\alpha}}_{\mathrm{MLE}} \mid y)$, while the Dirichlet-projected feature $\hat{\boldsymbol{\alpha}}_{\mathrm{MLE}}$ itself remains purely trial-specific. The final calibrated prediction is then obtained via $\hat{y}_{\mathrm{cal}} = \arg\max_{y \in \mathcal{L}} p_{\mathrm{cal}}(y \mid \hat{\boldsymbol{\alpha}}_{\mathrm{MLE}})$. Subsequently, the Dirichlet parameter $\hat{\boldsymbol{\alpha}}_{\mathrm{MLE}}$ of the current test EEG trial is updated into the memory bank according to its confidence and predicted label. We maintain a fixed-size memory bank and discard the oldest entries when full, while only inserting trials that satisfy both a confidence threshold $\tau_{\mathrm{conf}}$ and an entropy threshold $\tau_{\mathrm{ent}}$. In this way, highly uncertain or noisy recent trials are prevented from entering the memory, very old trials are gradually forgotten. Each inserted trial contributes at most $1/M$ to the GMM fitting, so a small set of recent trials cannot dominate the GMM posterior fusion. After each insertion, we refit the class-specific GMM on the corresponding memory bank using standard Expectation–Maximization (EM). This is feasible because the Dirichlet features are very low-dimensional, so full EM refitting incurs only small overhead, whereas more sophisticated incremental or online clustering would introduce unnecessary complexity without empirical benefits in our setting.

Algorithm 1 summarizes the pseudo-code of the proposed BTTA-DG framework (see Appendix D).

$$M_{\hat{y}_{\mathrm{cal}}} \leftarrow M_{\hat{y}_{\mathrm{cal}}} \cup \{\hat{\boldsymbol{\alpha}}_{\mathrm{MLE}}\}. \tag{7}$$

3) THEORETICAL ANALYSIS FOR ENHANCED TEST-TIME ADAPTABILITY

The central innovation of our work is the shift from adapting on point estimates (i.e., pseudo-labels) to adapting on probabilistic distributional representations. Conventional TTA methods are highly sensitive to the noise and uncertainty inherent in single predictions from a domain-shifted model. Furthermore, shallow alignment techniques fail to capture how the complex space of temporal predictive embeddings deforms in a new domain. Our key insight is that the domain shift is more reliably expressed in the distribution of the model's sequential predictions rather than in any single prediction.

Instead of directly using the model's output point estimates, we introduce Dirichlet feature projection to model the entire distribution of the sequential categorical embeddings for each trial. This yields a low-dimensional Dirichlet parameter vector $\boldsymbol{\alpha}$. The vector $\boldsymbol{\alpha}$ provides a richer representation of a trial's predictive characteristics than a simple class prediction. Each component $\alpha_i$ reflects the "concentration" or evidence for class $i$, while the total scale $\alpha_0$ relates to the predictive uncertainty or variance throughout the trial. When encountering a new domain, inherent signal differences cause

shifts in these predictive distributions. Our Dirichlet projection explicitly captures this distributional shift in a compact parametric form. This is a more robust and informative feature for adaptation, as it encodes not just what the model predicts, but also how confident and consistent that prediction is across time, which is crucial for characterizing a new domain.

This principled representation is what enables high test-time adaptability. By capturing the essence of the domain shift in a low-dimensional parameter space, we can perform effective calibration using well-established density estimation (GMM) and inference (Bayesian) techniques. This completely bypasses the need for gradient-based optimization, thus circumventing catastrophic forgetting and computational inefficiency. The theoretical soundness is empirically validated through visualization analysis (Figure 3), demonstrating well-separated class clusters with high inter-class KL divergence (>31.85) and low intra-class covariance (<0.27), confirming successful mapping of domain-shifted signals into a discriminative latent space.

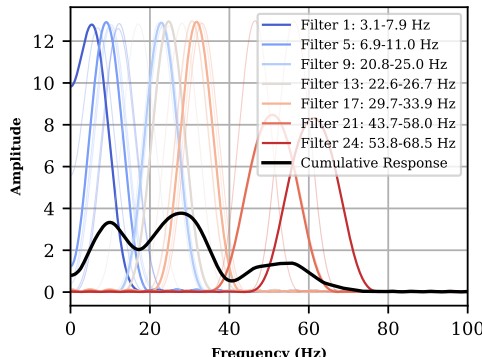

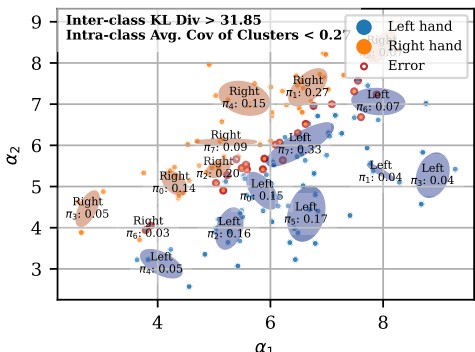

Figure 2: Frequency responses of the 24 learned SincConv filters (color-coded) and their cumulative response (black), which shows spectral energy concentration.

Figure 3: Scatter of Dirichlet parameter estimated for test EEG trials, color-coded by class with misclassified trials outlined in red. GMM ellipses reveal weight and covariance of clusters.

## 3 EXPERIMENTAL RESULTS

### 3.1 IMPLEMENTATION

**Datasets** The BTTA-DG framework was evaluated on three MOABB (Jayaram & Barachant, 2018) motor–imagery datasets BNCI2014001, BNCI2014002, and BNCI2015001, and SHU MI dataset (Ma et al., 2022). Key characteristics are summarized in Table 1. For BNCI2014001, only two classes (r/l hand imaginations) were used. For all datasets, a cross-subject leave-one-subject-out (LOSO) protocol was adopted, using only the first session from each dataset. The source model was pretrained on the training set, and during test-time adaptation, test trials arrived sequentially one-by-one in the online adaptation setting. Preprocessing included only a 1–48 Hz bandpass filter and Euclidean Alignment (He & Wu, 2019).

Table 1: Summary of the four MI EEG datasets

| Dataset | Number of subjects | Number of channels | Sampling rate (Hz) | Trial length (s) | Number of sessions | Trials in 1st session | Types of imaginations |
|---|---|---|---|---|---|---|---|
| BNCI2014001 | 9 | 22 | 250 | 4 | 2 | 288 | r/l hand, feet, tongue |
| BNCI2014002 | 14 | 15 | 512 | 5 | 1 | 160 | right hand, both feet |
| BNCI2015001 | 12 | 13 | 512 | 5 | 3 | 400 | right hand, both feet |
| SHU MI Dataset | 25 | 32 | 512 | 4 | 5 | 100 | left hand, right hand |
| BCI C IV 2b | 9 | 3 | 250 | 4 | 5 | 120 | left hand, right hand |

**Baselines** To comprehensively assess the performance of BTTA-DG, we compared it against traditional classification methods, Transformer-based method, optimal transport-based method and

Table 2: Cross-subject adaptation accuracy (%) on BNCI2014001, with an asterisk(*) denoting the significance level (*: p<0.05).

| Setting | Method | S0 | S1 | S2 | S3 | S4 | S5 | S6 | S7 | S8 | Avg. |
|---|---|---|---|---|---|---|---|---|---|---|---|
| Source | CSP | 83.33 | 52.08 | 97.92 | 75.00 | 56.25 | 67.36 | 72.22 | 88.19 | 71.53 | 73.77 |
| | EEGNet | 83.19 | 60.28 | 92.08 | 67.92 | 57.22 | 72.50 | 64.86 | 86.11 | 79.44 | $73.73_{\pm1.11}$ |
| | EEG conformer | 81.18 | 64.16 | 96.80 | 74.44 | 58.47 | 70.76 | 64.17 | 92.71 | 79.03 | $75.75_{\pm2.15}$ |
| | SincAdaptNet | 84.97 | 63.93 | 97.68 | 77.13 | 56.22 | 72.68 | 67.26 | 93.86 | 79.56 | $77.03_{\pm1.31}$ |
| Online TTA | BN-adapt | 84.97 | 63.93 | 97.68 | 77.13 | 56.22 | 72.68 | 67.26 | 93.86 | 79.56 | $77.03_{\pm1.31}$ |
| | Tent | 75.97 | 57.92 | 94.51 | 68.54 | 52.22 | 65.21 | 59.38 | 90.14 | 68.19 | $70.23_{\pm3.28}$ |
| | PL | 76.46 | 56.67 | 97.92 | 70.34 | 52.29 | 66.32 | 60.42 | 93.89 | 72.15 | $71.83_{\pm3.21}$ |
| | CoTTA | 85.00 | 63.68 | 98.05 | 76.32 | 57.22 | 72.08 | 67.64 | 94.63 | 80.48 | $77.24_{\pm1.51}$ |
| | SAR | 84.24 | 63.40 | 97.36 | 76.25 | 54.72 | 69.10 | 67.50 | 93.54 | 80.28 | $76.27_{\pm1.92}$ |
| | T-TIME | 84.44 | 61.94 | 97.43 | 76.11 | 56.60 | 69.38 | 63.13 | 94.65 | 79.38 | $75.90_{\pm1.95}$ |
| | OTTA | 84.43 | 63.60 | 97.14 | 77.63 | 57.63 | 73.04 | 66.44 | 95.26 | **83.14** | $77.58_{\pm1.33}$ |
| | BTTA-DG | **87.51*** | **66.67*** | **98.61*** | 77.08 | 57.64 | **73.61** | **68.75*** | **95.83*** | 82.64 | $\mathbf{78.70^*_{\pm1.32}}$ |

Table 3: Cross-subject adaptation accuracy (%) on BNCI2014002, with an asterisk(*) denoting the significance level (*: p<0.05).

| Setting | Method | S0 | S1 | S2 | S3 | S4 | S5 | S6 | S7 | S8 | S9 | S10 | S11 | S12 | S13 | Avg. |
|---|---|---|---|---|---|---|---|---|---|---|---|---|---|---|---|---|
| Source | CSP | 62.00 | 82.00 | 98.00 | 76.00 | 79.00 | 70.00 | 84.00 | 67.00 | 94.00 | 72.00 | 68.00 | 63.00 | 59.00 | 44.00 | 72.71 |
| | EEGNet | 65.00 | 80.00 | 83.00 | 80.20 | 74.20 | 68.20 | 88.80 | 54.60 | 91.20 | 75.00 | 81.00 | 72.00 | 59.80 | 51.40 | $73.17_{\pm0.59}$ |
| | EEG conformer | 66.50 | 80.50 | 95.00 | 78.10 | 80.10 | 70.30 | 90.50 | 70.20 | 92.40 | 73.00 | 79.50 | 75.80 | 58.20 | 47.30 | $75.53_{\pm1.85}$ |
| | SincAdaptNet | 67.90 | 80.10 | 99.00 | 79.30 | 81.00 | 72.20 | 93.10 | 76.10 | 94.00 | 75.90 | 81.20 | 80.90 | 60.40 | 51.60 | $78.05_{\pm2.48}$ |
| Online TTA | BN-adapt | 67.90 | 80.10 | 99.00 | 79.30 | 81.00 | 72.20 | 93.10 | 76.10 | 94.00 | 75.90 | 81.20 | 80.90 | 60.40 | 51.60 | $78.05_{\pm2.48}$ |
| | Tent | 57.20 | 70.40 | 90.30 | 70.70 | 67.10 | 68.30 | 88.10 | 64.30 | 91.60 | 64.10 | 77.10 | 70.40 | 52.30 | 49.20 | $70.08_{\pm4.30}$ |
| | PL | 57.70 | 72.30 | 99.80 | 73.80 | 71.60 | 67.70 | 92.10 | 65.80 | 93.60 | 64.80 | 78.70 | 68.40 | 53.10 | 49.20 | $72.04_{\pm3.54}$ |
| | CoTTA | 66.80 | 79.90 | 99.90 | 79.40 | 81.50 | 72.30 | 93.80 | 75.50 | **95.00** | 76.30 | 81.50 | 80.50 | 59.30 | 52.00 | $78.12_{\pm1.71}$ |
| | SAR | 66.20 | 80.10 | 99.60 | 77.20 | 80.80 | 72.70 | 91.90 | 74.40 | 94.40 | 74.10 | 81.30 | 79.40 | 57.00 | 49.10 | $77.02_{\pm2.03}$ |
| | T-TIME | 67.10 | 78.80 | 99.40 | 79.60 | 82.70 | 70.50 | 91.80 | 74.20 | 92.80 | 73.80 | 79.60 | 79.30 | 57.70 | 48.60 | $76.85_{\pm2.34}$ |
| | OTTA | 67.40 | **83.00** | 98.50 | 78.10 | 81.30 | 74.40 | 94.60 | 70.00 | 94.70 | **78.60** | 77.20 | 82.50 | **63.90** | 51.80 | $78.29_{\pm1.68}$ |
| | BTTA-DG | **69.00*** | **83.00** | **100.00** | **82.00*** | **83.00** | **75.00** | **95.00** | **80.00*** | **95.00** | 77.00 | **84.00*** | **83.00** | 63.40 | **54.60*** | $\mathbf{80.29_{\pm1.07}}$ |

state-of-the-art TTA methods, including CSP (Blankertz et al., 2007), EEGNet (Lawhern et al., 2018), EEG Conformer (Song et al., 2022), BN-adapt (Schneider et al., 2020), Tent (Wang et al., 2020), PL (Lee et al., 2013), SAR (Niu et al., 2023), CoTTA (Wang et al., 2022), T-TIME (Li et al., 2023), and OTTA (Wimpff et al., 2024). All experiments were run independently 10 times, and the average results are reported. Detailed experimental settings, including dataset descriptions, baseline methods, and hyperparameters, are provided in Appendix E.

## 3.2 MAIN RESULTS

Table 2 reports the cross-subject accuracies on BNCI2014001, including the source model and online TTA techniques. Our proposed BTTA-DG achieves state-of-the-art performance with an average accuracy of 78.70%. Notably, BTTA-DG excels in most subjects, highlighting its robust generalization capabilities across different subjects. We also observe that several gradient-based TTA baselines drop after applying TTA. Because in EEG's online single-trial adaptaion (batch size = 1), noisy trials induce misleading gradients that update BN weights and overwrite pre-trained structure—i.e., catastrophic forgetting. BTTA-DG avoids this failure mode by freezing the network and adapting in a probabilistic parameter space, yielding gradient-free calibration without destructive updates. Table 3 and Table 4 summarize LOSO cross-subject accuracies on BNCI2014002 and BNCI2015001. Results on SHU MI dataset and detailed statistical significance analysis are presented in Appendix F. Given the practical constraints of portable EEG hardware, we further evaluated BTTA-DG on BCI Competition IV 2b, which contains only three EEG channels. (Appendix G)

In addition, we also assessed the performance of BTTA-DG in a within-subject cross-session experiment. We pretrained the model on the first session and tested it on the second session of BNCI2014001, simulating session shifts in the same subject's motor imagery. The results of this cross-session adaptation are presented in Table 5, demonstrating that BTTA-DG effectively adapts to different sessions too (86.50% ± 2.49%). Statistical significance analysis are presented in Appendix H. Additionally, a sliding-window experiment on BNCI2014001 further shows that BTTA-DG also improves over the source model in online BCIs requiring low latency and continuous feedback. (Appendix I). Furthermore, to relate the cross-subject TTA gains to each subject's intrinsic "decodeability", we report three-fold within-subject k-fold adaptation accuracies on BNCI2014001 in Appendix J.

Table 4: Cross-subject adaptation accuracy (%) on BNCI2015001, with an asterisk(*) denoting the significance level (*: p<0.05).

| Setting | Method | S0 | S1 | S2 | S3 | S4 | S5 | S6 | S7 | S8 | S9 | S10 | S11 | Avg. |
|---|---|---|---|---|---|---|---|---|---|---|---|---|---|---|
| Source | CSP | 93.50 | 93.50 | 86.50 | 85.00 | 79.00 | 62.00 | 65.00 | 59.00 | 59.50 | 65.00 | 59.50 | 56.50 | 72.00 |
| | EEGNet | 91.50 | 95.00 | 75.70 | 85.90 | 81.30 | 68.60 | 65.20 | 64.30 | 63.00 | 66.50 | 57.50 | 55.20 | $72.48_{\pm0.52}$ |
| | EEG conformer | 94.00 | 95.20 | 83.10 | 86.50 | 84.80 | 66.40 | 68.50 | 64.10 | 63.50 | 65.00 | 58.30 | 56.40 | $73.82_{\pm1.65}$ |
| | SincAdaptNet | 97.35 | 94.70 | 90.70 | 87.65 | 87.30 | 64.10 | 72.30 | 65.30 | 64.75 | 63.15 | 59.80 | 58.70 | $75.48_{\pm1.84}$ |
| Online TTA | BN-adapt | 97.35 | 94.70 | 90.70 | 87.65 | 87.30 | 64.10 | 72.30 | 65.30 | 64.75 | 63.15 | 59.80 | 58.70 | $75.48_{\pm1.84}$ |
| | Tent | 80.30 | 68.65 | 73.75 | 68.05 | 68.80 | 55.15 | 56.60 | 57.15 | 54.45 | 55.55 | 51.60 | 50.25 | $61.69_{\pm4.33}$ |
| | PL | 97.85 | 90.50 | 77.30 | 80.10 | 70.30 | 56.30 | 58.00 | 56.75 | 54.25 | 55.70 | 51.60 | 49.90 | $66.55_{\pm4.17}$ |
| | CoTTA | 97.90 | 95.35 | 90.80 | 88.45 | 88.00 | 63.65 | 72.10 | 65.20 | 65.75 | 63.45 | 58.90 | 58.50 | $75.67_{\pm2.21}$ |
| | SAR | 96.85 | 87.75 | 90.60 | 73.30 | 87.65 | 64.60 | 72.65 | 64.80 | 65.25 | 62.75 | 60.70 | 57.45 | $73.70_{\pm3.32}$ |
| | T-TIME | 95.40 | 93.70 | 89.65 | 86.40 | 83.45 | 63.25 | 73.25 | 62.50 | 64.45 | 60.10 | 59.00 | 57.10 | $74.02_{\pm2.40}$ |
| | OTTA | **98.95** | 95.00 | 89.05 | 85.45 | 87.70 | **71.55** | 69.70 | **67.55** | 61.45 | 66.45 | 62.60 | 59.00 | $76.20_{\pm1.50}$ |
| | BTTA-DG | 98.50 | 96.00* | 92.00* | 90.80* | 88.50 | 65.50 | 75.50 | 67.50 | 65.75 | 65.50 | 64.50* | 65.00* | $\mathbf{77.92^*_{\pm1.76}}$ |

Table 5: Cross-session adaptation accuracy (%) on BNCI2014001, with an asterisk(*) denoting the significance level (*: p<0.05).

| Setting | Method | S0 | S1 | S2 | S3 | S4 | S5 | S6 | S7 | S8 | Avg. |
|---|---|---|---|---|---|---|---|---|---|---|---|
| Source | CSP | 88.19 | 54.17 | 97.22 | 65.97 | 48.61 | 70.14 | 68.06 | 94.44 | 90.97 | 75.31 |
| | EEGNet | 86.81 | **63.54** | 94.65 | 70.97 | 72.92 | 68.61 | 73.26 | 93.47 | 92.71 | $79.66_{\pm2.52}$ |
| | EEG Conformer | 87.92 | 58.75 | 97.15 | 70.35 | 75.83 | 68.54 | 77.77 | 95.63 | 89.23 | $80.13_{\pm3.18}$ |
| | SincAdaptNet | 84.69 | 56.64 | 98.51 | 70.32 | 86.22 | 72.06 | 82.47 | 97.54 | 92.47 | $82.33_{\pm2.62}$ |
| Online TTA | BN-adapt | 84.69 | 56.64 | 98.51 | 70.32 | 86.22 | 72.06 | 82.47 | 97.54 | 92.47 | $82.33_{\pm2.62}$ |
| | Tent | 80.35 | 51.18 | 99.03 | 57.92 | 64.38 | 62.22 | 64.51 | 93.06 | 90.83 | $73.72_{\pm4.77}$ |
| | PL | 77.71 | 51.74 | 98.75 | 58.13 | 75.28 | 64.58 | 68.75 | 97.01 | 91.46 | $75.93_{\pm4.72}$ |
| | CoTTA | 85.63 | 54.51 | 99.44 | 69.03 | 86.46 | 72.36 | 82.71 | 98.33 | 93.06 | $82.39_{\pm2.83}$ |
| | SAR | 86.32 | 55.00 | 99.24 | 71.11 | 86.11 | 70.49 | 82.92 | 96.53 | 91.88 | $82.18_{\pm2.99}$ |
| | T-TIME | 78.40 | 54.03 | 98.33 | 69.79 | 81.94 | 70.49 | 80.69 | 97.50 | 91.53 | $80.30_{\pm3.42}$ |
| | OTTA | **89.71** | 55.89 | 96.79 | 72.42 | 91.58 | 73.67 | 87.49 | 96.03 | 91.58 | $83.91_{\pm2.25}$ |
| | BTTA-DG | 85.42 | 62.50 | 100.00* | 76.39* | 91.67 | 77.78* | 90.97 | 100.00* | 93.75 | $\mathbf{86.50^*_{\pm2.49}}$ |

Table 6: Quantitative alignment of learned Sinc filter passbands with known MI-EEG rhythms on BNCI2014001. Values denote, for each subject, the percentage (%) of filters whose passbands overlap each band.

| Frequency band | S0 | S1 | S2 | S3 | S4 | S5 | S6 | S7 | S8 | Avg. |
|---|---|---|---|---|---|---|---|---|---|---|
| $\mu$ (8–13 Hz) | 24.17 | 25.83 | 25.83 | 23.33 | 24.17 | 27.50 | 24.17 | 24.17 | 25.00 | 24.91 |
| $\beta$ (13–30 Hz) | 32.50 | 35.00 | 28.33 | 28.33 | 43.33 | 25.83 | 30.00 | 28.33 | 29.17 | 31.20 |
| $\gamma$ (30–45 Hz) | 35.83 | 30.83 | 34.17 | 40.00 | 26.67 | 40.00 | 40.00 | 40.00 | 36.67 | 36.02 |
| Other | 7.50 | 8.33 | 11.67 | 8.33 | 5.83 | 6.67 | 5.83 | 7.50 | 9.17 | 7.87 |

To verify that SincAdaptNet's Sinc-Conv layer learns meaningful bandpass filters, Figure 2 illustrates the frequency responses of each learned filter on BNCI2014001 and their cumulative response in cross-subject LOSO setting (training set: S1–S8; test set: S0). Learned filters partition into three physiologically meaningful ranges – mu rhythm (8-13 Hz, Filters 1-7), beta rhythm (13-30 Hz, Filters 8-17), and gamma rhythm (30-68.5 Hz, Filters 18-24). Cumulative response (black curve) peaks in the mu (11.2 Hz), beta (30.5 Hz), and gamma (55.3 Hz) bands, demonstrating significant energy concentration of MI EEG. Similar spectral patterns emerge across different test subjects (see Appendix K). Beyond these qualitative plots, Table 6 summarizes a quantitative alignment analysis on BNCI2014001 (see Appendix L for calculation details). On average, about 25% of filters align with the $\mu$ band, 31% with the $\beta$ band, and 36% with the low-$\gamma$ band, leaving fewer than 8% outside these MI rhythms. This concentration on well-established sensorimotor frequencies further supports the physiological interpretability of the SincAdaptNet front-end.

Figure 14 (in Appendix M) presents the 16 spatial kernels learned by SincAdaptNet's Spat-Conv layer as scalp topographies, each displaying a distinct electrode weighting akin to those produced by the CSP (Blankertz et al., 2007). The range of frontal, central, parietal, and occipital patterns (Decety, 1996; Lesser et al., 1998) highlights the model's capacity to learn multiple spatial representations of EEG activity.

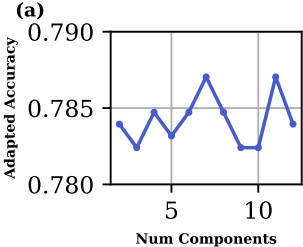 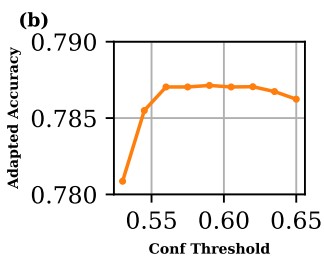 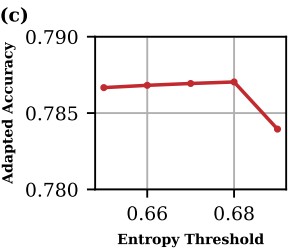

Figure 4: Sensitivity of BTTA-DG to key hyperparameters.

Figure 3 visualizes the low-dimensional Dirichlet parameter estimated from test EEG trials and GMM clustering outcomes on BNCI2014001 in cross-subject LOSO setting (training set: S1–S8; test set: S0). The low-dimensional Dirichlet parameter of EEG representation exhibits good class separability, with misclassified samples predominantly located near class boundaries. Moreover, a high inter-class KL divergence (31.85) and a low intra-cluster covariance (0.27) indicate that GMM effectively captures both global and local information of Dirichlet parameter distribution, providing a robust statistical foundation for Bayesian inference. The geometry of Dirichlet parameters and cluster remains stable for other test subjects (see Appendix N).

**Computational efficiency**    To assess the computational efficiency of BTTA-DG, we measured the average inference time and the number of floating-point operations (FLOPs) per trial for each method on the BNCI2014001 dataset. As presented in Table 7, BTTA-DG achieves real-time performance with an average inference time of 15.7 ms per trial – 17.8% faster than T-TIME (Li et al., 2023) and 24.2% faster than OTTA (Wimpff et al., 2024), which are the recent baselines for EEG-TTA. BN-adapt is the fastest method because it merely recomputes batch-normalization statistics without any gradient-based optimization or probabilistic inference. Detailed computational complexity analysis and memory usage are in Appendix O.

**Ablation study**    To dissect the contributions of our proposed components, we conducted an ablation study, with the results summarized in Table 8. We start with the **SincAdaptNet (Source Only)** model as our baseline. We then incorporate **Euclidean Alignment (+ EA)** to establish the performance of a standard domain alignment technique. We further evaluate **BTTA-DG w/o EA**, which removes EA but retains the Dirichlet feature projection and GMM-driven inference. We also include **SincAdaptNet + EA + GMM**, which performs GMM-driven inference directly on the time-averaged class-probability vector. The crucial next step isolates the effectiveness of our Dirichlet projection by applying the **Dirichlet Projection without the GMM-driven inference (+ Dirichlet Projection)**, instead using a simple classifier on the projected parameters. Finally, our **BTTA-DG (Full Model)** integrates all components. Across all datasets and settings, EA provides an initial improvement over SincAdaptNet, and BTTA-DG w/o EA also yields gains over the source-only model, indicating that the Dirichlet+GMM module has standalone adaptation capability. Introducing GMM calibration on the mean class-probability (+ EA + GMM) leads to marginal improvements over + EA. Consistently, SincAdaptNet + EA + Dirichlet outperforms both + EA and + EA + GMM, while the full BTTA-DG, which combines EA, Dirichlet projection and GMM-based Bayesian inference, achieves the best performance in every setting, with absolute gains of roughly 2–6% over the source-only baseline.

**Sensitivity analysis**    To assess the robustness of BTTA-DG, sensitivity analyzes were conducted on three key hyperparameters: the number of GMM components $K$, the minimum confidence threshold $\tau_{\text{conf}}$ and the maximum entropy threshold $\tau_{\text{ent}}$. $\tau_{\text{conf}}$ and $\tau_{\text{ent}}$ govern the conditions under which Dirichlet parameters of test trials are stored in the memory bank: Dirichlet parameters are retained if the confidence of trials exceeds $\tau_{\text{conf}}$ and their entropy is below $\tau_{\text{ent}}$, ensuring that only high-certainty, low-uncertainty test data contribute to the adaptation process. As shown in Figure 4, BTTA-DG exhibits robustness across parameter variations. Accuracy remains stable (78.2%–78.7%) for $K \in [2, 12]$. Higher thresholds (0.53–0.65) improve accuracy from 78.0% to 78.7%, filtering low-confidence trials to reduce noise in adaptation. Low entropy (0.65–0.70) sustains accuracy at 78.0%–78.7%, ensuring only high-certainty trials enter the memory bank.

**Sensitivity to Online Class Imbalance**   To evaluate the robustness of BTTA-DG in practical scenarios where data flow may not be uniformly distributed, we conducted an experiment on the BNCI2014001 dataset under artificially induced online class imbalance. We systematically varied the class ratio in the test set from a balanced 1:1 distribution to a severely imbalanced 1:0.25. Table 9 reports the class-wise and overall accuracies under these ratios. While the overall accuracy gracefully degrades with increasing imbalance, we observe that the accuracy for the minority class (Class 1) conversely improves as its prevalence decreases. These results indicate that BTTA-DG does not collapse under imbalance but instead specializes to rare events, which is desirable for many real-world BCI scenarios. A more detailed interpretation is provided in Appendix P.

Table 7: Average inference time and FLOPs per trial on BNCI2014001.

| Method | BN-adapt | Tent | PL | CoTTA | SAR | T-TIME | OTTA | BTTA-DG |
|---|---|---|---|---|---|---|---|---|
| **Time (ms)** | 5.1 | 18.4 | 17.8 | 23.0 | 32.5 | 18.5 | 19.5 | 15.7 |
| **FLOPs (M / trial)** | 133.3 | 266.7 | 266.7 | 333.4 | 466.9 | 266.7 | 330.5 | 141.6 |

Table 8: Ablation Study across settings and MI datasets. Mean $\pm$ s.d. accuracy (%).

| Method | BNCI2014001 | | BNCI2014002 | BNCI2015001 | SHU MI |
|---|---|---|---|---|---|
| | cross-session | cross-subject | cross-subject | cross-subject | cross-subject |
| SincAdaptNet (Source Only) | $80.62 \pm 2.70$ | $75.30 \pm 1.82$ | $76.40 \pm 1.62$ | $73.92 \pm 1.95$ | $61.02 \pm 1.70$ |
| BTTA-DG w/o EA | $81.88 \pm 2.58$ | $76.85 \pm 1.65$ | $77.55 \pm 1.59$ | $75.06 \pm 1.88$ | $61.90 \pm 1.68$ |
| SincAdaptNet + EA | $82.33 \pm 2.62$ | $77.03 \pm 1.31$ | $78.05 \pm 2.48$ | $75.48 \pm 1.84$ | $62.42 \pm 1.72$ |
| SincAdaptNet + EA + GMM | $82.47 \pm 2.59$ | $77.55 \pm 1.35$ | $78.25 \pm 2.36$ | $75.62 \pm 1.82$ | $62.58 \pm 1.70$ |
| SincAdaptNet + EA + Dirichlet | $84.04 \pm 2.55$ | $77.61 \pm 1.43$ | $78.88 \pm 1.47$ | $76.36 \pm 1.78$ | $63.24 \pm 1.78$ |
| **BTTA-DG (Full Model)** | $\mathbf{86.50 \pm 2.49}$ | $\mathbf{78.70 \pm 1.32}$ | $\mathbf{80.29 \pm 1.07}$ | $\mathbf{77.92 \pm 1.76}$ | $\mathbf{64.06 \pm 1.92}$ |

Table 9: Performance of BTTA-DG under varying online class imbalance ratios on the BNCI2014001 dataset. As imbalance increases, the model specializes, improving minority class accuracy.

| Class Ratio (0 : 1) | Accuracy Class 0 (%) | Accuracy Class 1 (%) | Overall Accuracy (%) |
|---|---|---|---|
| 1 : 1 | $77.01 \pm 1.53$ | $80.40 \pm 1.45$ | $\mathbf{78.70 \pm 1.32}$ |
| 1 : 0.75 | $74.07 \pm 1.44$ | $80.45 \pm 1.34$ | $76.81 \pm 1.25$ |
| 1 : 0.5 | $69.75 \pm 1.52$ | $83.02 \pm 1.42$ | $74.17 \pm 1.28$ |
| 1 : 0.25 | $64.67 \pm 1.18$ | $85.19 \pm 1.64$ | $68.77 \pm 1.19$ |

## 4   CONCLUSION

In this paper, we presented BTTA-DG, a novel gradient-free and efficient TTA framework for MI-EEG decoding. By projecting temporal predictive embeddings into a compact Dirichlet parameter space, our method captures predictive uncertainty and models the target domain's feature distribution. A GMM is employed to summarize historical Dirichlet parameters, preserving global and neighborhood information of test trials. Subsequent Bayesian inference integrates learned historical evidence with the network's priors, effectively bridging domain gaps without the risk of catastrophic forgetting. Our extensive experiments validate that BTTA-DG achieves state-of-the-art adaptation performance, significantly outperforming existing EEG-TTA methods while operating at real-time speeds. Although density estimation on the memory buffer benefits from reasonably balanced EEG data, our online class-imbalance study indicates that BTTA-DG remains robust even under imbalanced class ratios. The explicit handling more extreme or non-stationary imbalance is an interesting direction for future work. Beyond MI-EEG, the same principle of Dirichlet feature projection plus GMM-based Bayesian calibration could in principle be transferred to other neural modalities (e.g., fNIRS, ECoG) and to broader BCI settings. In future work, we also commit to integrate BTTA-DG with large pretrained EEG models to explore whether the gains are additive. By enabling high-performance adaptation without requiring new labeled data, our framework reduces user burden and represents a significant step towards the development of practical, real-world BCI systems.

ETHICS STATEMENT

The authors adhere to the ICLR Code of Ethics. The datasets utilized in this research are publicly available for all researchers in Brain-Computer Interface. No new data was collected from human subjects for this study. The original data collection was conducted by its respective creators under appropriate ethical protocols, including institutional review board (IRB) approval and informed consent from all participants. Our work is confined to the analysis of this existing, anonymized data and does not introduce new ethical concerns regarding human subjects.

REPRODUCIBILITY STATEMENT

We are committed to ensuring the reproducibility of our research. All experiments were conducted on the publicly accessible datasets. Comprehensive implementation details, including data preprocessing steps, model architecture, optimizer settings, and the full set of hyperparameters along with the strategy for their selection, are thoroughly documented in Section 3.1 and Appendix E.

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

# A    RELATED WORK

Recent years have witnessed remarkable progress in large-scale EEG pretrained models trained on massive datasets. Representative models include BENDR (Kostas et al., 2021), BIOT (Yang et al., 2023), LaBraM (Jiang et al., 2024), EEGPT (Wang et al., 2024), CBraMod (Wang et al., 2025) and CoMET (Li et al., 2025). These foundation models demonstrate impressive capabilities in learning general representations across diverse EEG paradigms. However, their effectiveness in MI decoding remains limited despite their scale. As shown in Table 10, lightweight architectures achieve competitive or superior performance compared to large-scale models while being orders of magnitude more parameter-efficient. For instance, SincAdaptNet matches or surpasses all considered large-scale models, and it trails EEGPT by less than 1% on BCIC-IV-2a while outperforming it on BCIC-2b, despite using roughly four orders of magnitude fewer parameters.

Table 10: Comparison of large-scale pretrained EEG models and lightweight models on BCIC-IV-2a-4class and BCIC-2b (LOSO).

| Scale | Methods | Number of Params (M) | BCIC-IV-2a-4class | BCIC-2b |
|---|---|---|---|---|
| Large-scale foundation models | BENDR | 4.0 | 0.4899 | 0.7067 |
| | BIOT | 3.2 | 0.4590 | 0.6409 |
| | LaBraM | 5.8 | 0.5613 | 0.6851 |
| | EEGPT | 25.0 | 0.5846 | 0.7212 |
| | CBraMod | 4.0 | 0.5585 | 0.6735 |
| Lightweight models | EEGNet | 0.003 | 0.5685 | 0.7429 |
| | EEGConformer | 0.55 | 0.5341 | 0.7361 |
| | SincAdaptNet | 0.0015 | 0.5764 | 0.7632 |

This performance gap stems from two fundamental challenges. First, MI-induced EEG signals are inherently weak and highly variable compared to other paradigms like visual evoked potentials. Unlike P300 or SSVEP which show consistent patterns across subjects, MI relies on subjective imagination to trigger sensorimotor rhythm ERD/ERS. This results in subtle power changes in specific bands (e.g. $\mu$: 8–13 Hz, $\beta$: 18–30 Hz) that are often masked by noise. Moreover, these subtle power changes vary significantly across subjects. Some subjects demonstrate clear $\mu$-rhythm ERD/ERS while others show only minimal changes, which is the "BCI illiteracy" phenomenon Dreyer et al. (2023). This extreme weakness and variability make unified feature extraction challenging. Large-scale models may not capture these subtle subject-specific nuances without task-aligned pretraining.

Second, current pre-training strategies for EEG foundation models may face limitations. Most foundation models are trained with objectives like masked reconstruction or contrastive learning that are inherited from NLP and vision tasks, to learn general representations from unlabeled EEG data Yuxuan et al. (2025). However, these objectives may not align with task-specific requirements. For instance, masked reconstruction encourages the model to recover missing signal segments based on contextual patterns. These patterns may emphasize global temporal structure that relates to noise rather than task-relevant features. In addition, many models mix diverse data without task/subject differentiation Chen et al. (2025), leading to "averaged" representations of unclear utility. This explains why simple linear classifiers perform poorly on these pretrained representations in certain case, where the features lack linearly separable task-relevant information Yang et al. (2026). Consequently, they require substantial non-linear adaptation during fine-tuning, which can negate the benefits of pretraining and risk overfitting on limited downstream datasets.

These limitations highlight the need for specialized adaptation approaches that can handle MI's inherent variability. Online test-time adaptation (TTA) methods adapt source models and make simultaneous predictions during inference, utilizing unlabeled online target data (Liang et al., 2025; Xiao & Snoek, 2024). Over the past year, TTA methods have been extended from computer vision to MI EEG signal decoding.

Traditional TTA techniques (batch normalization calibration (Zhao et al., 2023), entropy minimization (Wang et al., 2020; Niu et al., 2023), pseudo-labeling (Lee et al., 2013; Iwasawa & Matsuo, 2021) and consistency regularization(Brahma & Rai, 2023)) can be broadly categorized into parameter

finetuning and non-finetuning methods. Non-finetuning methods adjust domain-specific statistics without gradient backpropagation, offering high computational efficiency but limited adaptability. For example, BN-Adapt (Schneider et al., 2020) recalculated BN layer statistics in target domain to mitigate distribution shifts. Parameter finetuning methods, on the other hand, include partial and full finetuning. Partial finetuning methods update only a subset of the network's parameters to balance adaptability and efficiency. For instance, Tent (Wang et al., 2020) minimized entropy by updating only the BN affine parameters. SAR (Niu et al., 2023) further introduced sharpness-aware entropy minimization technique to suppress noisy test samples with large gradients, stabilizing TTA. Full finetuning methods update the entire network using losses calculated by pseudo-labels. For example, Pseudo-Label (Lee et al., 2013) employed the model's high-confidence predictions as pseudo-labels for self-training, whereas CoTTA (Wang et al., 2022) utilized data augmentation to generate pseudo-labels and incorporates weight stochastic restoration to alleviate error accumulation and forgetting.

In MI decoding domain, to address the complexity of EEG signals, state-of-the-art MI-TTA frameworks integrate data alignment techniques with multiple parameter finetuning techniques. Specifically, OTTA (Wimpff et al., 2024) integrated Euclidean Alignment (EA) (Zanini et al., 2017) or Riemannian Alignment (RA) (He & Wu, 2019) techniques with entropy minimization of BN finetuning to reduce cross-subject domain shifts. MI-FTTA (Peng et al., 2025) combines teacher-student mutual learning and time-constrained sample selection to filter noisy pseudo labels, and employed BN statistics recalculation and prototype-based contrastive learning to enhance adaptation performance. T-TIME (Li et al., 2023) used ensemble learning for label prediction and finetuned classifiers by conditional entropy minimization and adaptive marginal distribution regularization, achieving higher performance. Latent alignment method (Bakas et al., 2025) introduced deep sets to EEG decoding and aligned distributions in the deep learning model's feature space. For large-scale online MI decoding, continual finetuning strategies (Wimpff et al., 2025) combined with TTA methods have been also explored to improve BCI performance in real-world applications. In addition to these unsupervised TTA approaches, Sartzetaki et al. (Sartzetaki et al., 2023) present a comprehensive multi-dataset study of fine-tuning, showing that even a small amount of labeled target data can boost MI decoding performance and highlighting the potential of supervised adaptation when such labels are available.

Despite the advancements, critical challenges persist. 1) Computational efficiency: Full finetuning methods incur high cost due to gradient updates, limiting real-time applicability. 2) Catastrophic forgetting: Continuous parameter updates risk overwriting pre-trained knowledge for MI decoding. 3) Shallow statistics-based adaptation: Methods lack theoretical modeling of deep feature shifts across domains. Therefore, developing lightweight, robust, and theoretically grounded MI-TTA methods remains a challenge.

# B    SINCADAPTNET DETAILS

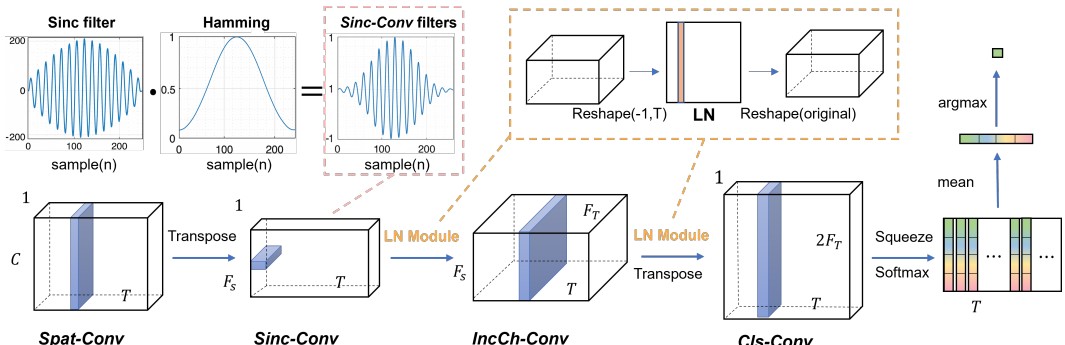

Figure 5: Architecture of SincAdaptNet. A compact four-layer encoder for MI EEG: Spat-Conv (spatial filtering) → Sinc-Conv (learnable, sinc-parameterized band-pass) → IncCh-Conv (channel expansion) → Cls-Conv (per-time class embeddings). Layer normalization is applied after temporal filtering and channel expansion to avoid reliance on batch statistics in online inference.

SincAdaptNet has only four convolutional layers. From a statistical perspective, MI decoding is largely governed by low-dimensional structure in the time–frequency domain (e.g., $\mu/\beta$ rhythms over a small set of motor channels), so a very deep architecture is more likely to increase variance than to capture additional task-relevant structure under limited MI-EEG data. SincAdaptNet imposes a strong inductive bias by parameterizing each temporal filter with only two spectral parameters (low cut-off and bandwidth), favoring band-limited solutions aligned with known MI neurophysiology and yielding a favorable bias–variance trade-off. Empirically, this lightweight encoder does not weaken representation capacity. Across all datasets in our experiments, SincAdaptNet matches or outperforms deeper backbones such as EEG Conformer while remaining computationally efficient for online TTA.

Moreover, the explicit frequency-band extraction of the Sinc-Conv layer directly benefits our Bayesian TTA framework. The learnable Sinc filters isolate physiologically critical $\mu$ and $\beta$ bands and suppress irrelevant spectral noise in raw EEG, producing a more stable and less ambiguous sequence of predictive embeddings over each trial. This cleaner input leads to more robust fixed-point Dirichlet estimation, improving the quality of the probabilistic representation that our GMM-based TTA module adapts on, and thereby enhancing robustness to non-stationarity in realistic online BCI scenarios while keeping the overall system efficient and interpretable.

The Sinc-Conv layer is not a standard convolution but is parametrically defined by the sinc function with learnable cutoff frequencies, making it highly interpretable. Full hyperparameter settings for the network are detailed in Appendix E. We have created an architecture diagram to visually clarify the network structure (Figure 5). Besides, we added the following SincAdaptNet architecture equations. Let $s \in \mathbb{R}^{C \times T}$ be an input EEG trial with $C$ channels and $T$ time points. The four-layer SincAdaptNet processes the input as follows:

**Spat-Conv**    Spatial filtering with $F_{\text{spat}}$ filters of size $(C \times 1)$: $s_{\text{spat}} = W_{\text{spat}} * s$ , where $W_{\text{spat}} \in \mathbb{R}^{F_{\text{spat}} \times C \times 1}$ are the spatial convolution kernels.

**Sinc-Conv**    Adaptive bandpass filtering with $F_{\text{sinc}}$ sinc-based filters: $s_{\text{sinc}} = h_{\text{sinc}} * s_{\text{spat}}$. The sinc kernel $h_{\text{sinc}}$ is parametrically generated from learnable cutoff frequencies $f_{\text{low}}$ and bandwidth $f_{\text{band}}$, with $f_{\text{high}} = f_{\text{low}} + f_{\text{band}}$. The frequency response of the ideal bandpass filter is:

$$H(f) = \text{rect}\left(\frac{f}{2f_{\text{high}}}\right) - \text{rect}\left(\frac{f}{2f_{\text{low}}}\right) \tag{8}$$

The time-domain impulse response is derived via the inverse Fourier transform,

$$h(n) = 2f_{\text{high}} \text{sinc}(2\pi f_{\text{high}} n) - 2f_{\text{low}} \text{sinc}(2\pi f_{\text{low}} n), \quad -\infty < n < \infty. \tag{9}$$

Since this ideal response is infinitely long, it is truncated and a Hamming window $w(n)$ is applied to mitigate spectral leakage, resulting in the convolution kernel,

$$h_{\text{windowed}}(n) = h(n) \cdot w(n), \quad -\frac{N_{\text{sinc}}}{2f_{\text{s}}} < n < \frac{N_{\text{sinc}}}{2f_{\text{s}}}, \tag{10}$$

where $N_{\text{sinc}}$ denotes the truncated length and $f_{\text{s}}$ is the sampling frequency. The *Sinc-Conv* layer comprises $F_{\text{sinc}}$ adaptive bandpass filters of size $(1 \times N_{\text{sinc}})$ and uses "SAME" padding to preserve temporal dimension.

**IncCh-Conv** Channel expansion with $2F_{\text{sinc}}$ filters of size $(F_{\text{spat}} \times 1)$: $s_{\text{inc}} = W_{\text{inc}} * s_{\text{sinc}}$, where $W_{\text{inc}} \in \mathbb{R}^{2F_{\text{sinc}} \times F_{\text{spat}} \times 1}$.

**Cls-Conv** Final layer with $|\mathcal{L}|$ filters: $X = W_{\text{cls}} * s_{\text{inc}}$, where $W_{\text{cls}} \in \mathbb{R}^{|\mathcal{L}| \times 2F_{\text{sinc}} \times 1}$, yielding output $X \in \mathbb{R}^{|\mathcal{L}| \times T}$. Each column is passed through a softmax to form a time-varying categorical vector, and the temporal average provides a robust prior for Bayesian calibration in our TTA framework.

## C  DIRICHLET PARAMETER ESTIMATION DETAILS

This section details the efficient algorithm used to compute the Maximum Likelihood Estimate (MLE) of the Dirichlet parameters $\alpha$ from a sequence of categorical probability vectors $X = [x_1, \ldots, x_T]$ with $x_j \in \Delta^{|\mathcal{L}|-1}$. The method is based on the well-established fixed-point iteration algorithm by Minka (2012), which we implement for our online TTA framework.

**Fixed-point iteration for MLE** Let $X = [x_1, x_2, \ldots, x_T]$ be a set of $T$ samples, where each $x_j = (x_{1j}, x_{2j}, \ldots, x_{|\mathcal{L}|j})^\top$ is a categorical probability vector sampled from a Dirichlet distribution with parameters $\alpha = (\alpha_1, \ldots, \alpha_{|\mathcal{L}|})$. The goal is to estimate $\mathcal{P}(X) = \hat{\alpha}_{\text{MLE}}$, the MLE of $\alpha$. The log-likelihood function for the Dirichlet distribution is employed.

$$\mathcal{L}(\alpha) = \sum_{j=1}^{T} \log \mathcal{D}(x_j; \alpha). \tag{11}$$

The maximum likelihood estimator $\hat{\alpha}_{\text{MLE}}$ can be obtained by iteratively computing

$$\alpha_i^{\text{new}} = \psi^{-1}\left(\psi(\alpha_0^{\text{old}}) + \frac{1}{T}\sum_{j=1}^{T} \log x_{ij}\right). \tag{12}$$

where $\psi(u) = \frac{d}{du}\Gamma(u)$ denotes the Digamma function.

Thus, if the inverse Digamma function $\psi^{-1}$ can be obtained, the iterative rule (12) is used to estimate the Dirichlet parameters for temporal predictive embeddings of each test EEG trial.

To accelerate convergence, we initialize the Dirichlet parameters $\alpha^{\text{init}}$ using the empirical mean $\bar{x}_i$ and variance $\sigma_i^2$ of the categorical probability vectors from temporal predictive embeddings. From the properties of the Dirichlet distribution, the mean and variance satisfy:

$$\bar{x}_i = \frac{\sum_{j=1}^{T} x_{ij}}{T} = \frac{\alpha_i}{\alpha_0}, \tag{13}$$

$$\sigma_i^2 = \frac{1}{T}\sum_{j=1}^{T}(x_{ij} - \bar{x}_i)^2 = \frac{\bar{x}_i(1 - \bar{x}_i)}{\alpha_0 + 1}. \tag{14}$$

Therefore, an initial estimate $\alpha^{\text{init}}$ can be computed as,

$$\alpha_0^{\text{init}} \approx \frac{1}{|\mathcal{L}|}\sum_{i=1}^{|\mathcal{L}|}\left(\frac{\bar{x}_i(1 - \bar{x}_i)}{\sigma_i^2} - 1\right), \tag{15}$$

$$\alpha_i^{\text{init}} = \alpha_0^{\text{init}}\bar{x}_i. \tag{16}$$

By iterating rule (12) with these initial estimates and solving for $\psi^{-1}$ via Newton's method (described below), we efficiently converge to the Dirichlet parameters $\hat{\alpha}_{\text{MLE}}$.

**Newton's iteration for calculating inverse Digamma**   The Digamma function is defined by

$$\psi\left(u\right) = \frac{d}{du}\Gamma\left(u\right) = -\gamma + \sum_{n=0}^{\infty}\left(\frac{1}{n+1} - \frac{1}{n+u}\right), \tag{17}$$

where Euler constant $\gamma = -\psi\left(1\right)$. To solve for $u = \psi^{-1}(v)$ from the equation $\psi(u) - v = 0$, we employ Newton's iteration (Gil et al., 2007). Given $v = \psi(\alpha_0^{\text{old}}) + \frac{1}{T}\sum_{j=1}^{T}\log x_{ij}$ in rule (12), the Newton iteration is formulated as,

$$u^{\text{new}} = u^{\text{old}} - \frac{\psi(u^{\text{old}}) - v}{\psi'(u^{\text{old}})}, \tag{18}$$

where $\psi'(u)$ denotes the Trigamma function.

To further accelerate convergence of Newton's iteration, the initial value $u^{\text{init}}$ is initialized using an approximate inverse function $\widetilde{\psi}^{-1}\left(v\right)$, where

$$\psi(u) = \widetilde{\psi}(u) = \begin{cases} \log\left(u - 1/2\right) & \text{if } u \geq 0.6, \\ -\frac{1}{u} - \gamma & \text{if } u < 0.6, \end{cases} \tag{19}$$

Yelds,

$$u^{\text{init}} = \widetilde{\psi}^{-1}\left(v\right) = \begin{cases} \exp\left(v + \frac{1}{2}\right), & \text{if } v \geq -2.22 \\ -\frac{1}{v+\gamma}, & \text{if } v < -2.22 \end{cases} \tag{20}$$

Once convergence is reached, we substitute $\psi^{-1}(v) = u^{\text{new}}$ into fixed-point iteration rule (12) to update $\alpha_i^{\text{new}} = u^{\text{new}}$. Iterating this process yields the estimated Dirichlet parameters $\hat{\alpha}_{\text{MLE}}$.

**Numerical stability.**   We clip $x_{ij}$ to $[10^{-7}, 1 - 10^{-7}]$ to avoid zeros, and early-stop when $\|\alpha^{\text{new}} - \alpha^{\text{old}}\|/\|\alpha^{\text{old}}\| < 10^{-4}$. In all datasets, the method converges within $O(I_{\text{iter}})$ iterations per trial, where $I_{\text{iter}}$ typically remains under 5–10.

# D    THE PSEUDO-CODE OF BTTA-DG

Algorithm 1 summarizes the pseudo-code of the proposed BTTA-DG framework.

---

**Algorithm 1:** BTTA-DG Framework

---

**Input:** Pretrained model $f_{\boldsymbol{\theta}}$, memory banks $M_y$, test trial $\boldsymbol{s}$, tolerance $\varepsilon$, $K$ GMM components, the minimum confidence threshold $\tau_{\text{conf}}$ and the maximum entropy threshold $\tau_{\text{ent}}$.
**Output:** Calibrated label $\hat{y}_{\text{cal}}$.

**// Feature extraction**
$\boldsymbol{X} \leftarrow g_{\text{enc}}(\boldsymbol{s})$
$p_{\boldsymbol{\theta}}(y) \leftarrow f_{\text{cls}}(\boldsymbol{X})$                                     // Obtain priori

**// Accelerated Dirichlet parameter estimation**
**// Initialize $\boldsymbol{\alpha}^{\text{init}}$**
**for** $i = 1$ **to** $|\mathcal{L}|$ **do**
$\quad$ $\bar{x}_i \leftarrow \frac{1}{T} \sum_{j=1}^{T} x_{ij}$
$\quad$ $\sigma_i^2 \leftarrow \frac{1}{T} \sum_{j=1}^{T} \left(x_{ij} - \bar{x}_i\right)^2$
$\quad$ $\alpha_0^{\text{init}} \leftarrow \frac{1}{|\mathcal{L}|} \sum_{i=1}^{|\mathcal{L}|} \left( \frac{\bar{x}_i(1-\bar{x}_i)}{\sigma_i^2} - 1 \right)$
$\quad$ $\alpha_i^{\text{init}} \leftarrow \alpha_0^{\text{init}} \bar{x}_i$
**end**
$\alpha_i^{\text{new}} \leftarrow \alpha_i^{\text{init}}, \alpha_0^{\text{new}} \leftarrow \alpha_0^{\text{init}}$

**// Fixed-point iteration for MLE**
**while** $\|\boldsymbol{\alpha}^{new} - \boldsymbol{\alpha}^{old}\| > \varepsilon$ **do**
$\quad$ $\boldsymbol{\alpha}^{\text{old}} \leftarrow \boldsymbol{\alpha}^{\text{new}}$
$\quad$ **for** $i = 1$ **to** $|\mathcal{L}|$ **do**
$\quad\quad$ $v \leftarrow \psi\left(\alpha_0^{\text{old}}\right) + \frac{1}{T} \sum_{j=1}^{T} \log x_{ij}$
$\quad\quad$ $u \leftarrow \text{NewtonSolver}(v, \varepsilon)$           // Using Equations (18), (19), and (20)
$\quad\quad$ $\alpha_i^{\text{new}} \leftarrow u$
$\quad$ **end**
$\quad$ $\alpha_0^{\text{new}} \leftarrow \sum_{i=1}^{|\mathcal{L}|} \alpha_i^{\text{new}}$
**end**
$\hat{\boldsymbol{\alpha}}_{\text{MLE}} \leftarrow \boldsymbol{\alpha}^{\text{new}}$

**// GMM-driven Bayesian inference**
**for** $y \in \mathcal{L}$ **do**
$\quad$ Fit GMM: $\{\pi_{y,k}, \boldsymbol{\mu}_{y,k}, \boldsymbol{\Sigma}_{y,k}\}_{k=1}^{K} \leftarrow \text{EM}(M_y, K)$
$\quad$ $p_{\text{GMM}}\left(\hat{\boldsymbol{\alpha}}_{\text{MLE}} \mid y\right) \leftarrow \sum_{k=1}^{K} \pi_{y,k} \mathcal{N}\left(\hat{\boldsymbol{\alpha}}_{\text{MLE}} \mid \boldsymbol{\mu}_{y,k}, \boldsymbol{\Sigma}_{y,k}\right)$
$\quad$ $p_{\text{cal}}\left(y \mid \hat{\boldsymbol{\alpha}}_{\text{MLE}}\right) \leftarrow \dfrac{p_{\text{GMM}}\left(\hat{\boldsymbol{\alpha}}_{\text{MLE}} \mid y\right) p_{\boldsymbol{\theta}}(y)}{\sum_{y'=1}^{|\mathcal{L}|} p_{\text{GMM}}\left(\hat{\boldsymbol{\alpha}}_{\text{MLE}} \mid y'\right) p_{\boldsymbol{\theta}}(y')}$
**end**

**// Final prediction and memory bank update**
$\hat{y}_{\text{cal}} \leftarrow \arg\max_{y \in \mathcal{L}} \; p_{\text{cal}}\left(y \mid \hat{\boldsymbol{\alpha}}_{\text{MLE}}\right)$
$\text{confidence} \leftarrow p_{\text{cal}}(\hat{y}_{\text{cal}} \mid \hat{\boldsymbol{\alpha}}_{\text{MLE}})$
$\text{entropy} \leftarrow -\sum_{y} p_{\text{cal}}(y \mid \hat{\boldsymbol{\alpha}}_{\text{MLE}}) \log p_{\text{cal}}(y \mid \hat{\boldsymbol{\alpha}}_{\text{MLE}})$
**if** $\text{confidence} \geq \tau_{conf}$ **and** $\text{entropy} \leq \tau_{ent}$ **then**
$\quad$ Drop oldest if $|M_{\hat{y}_{\text{cal}}}| \geq \text{buffer\_size}$
$\quad$ $M_{\hat{y}_{\text{cal}}} \leftarrow M_{\hat{y}_{\text{cal}}} \cup \{\hat{\boldsymbol{\alpha}}_{\text{MLE}}\}$
**end**
**return** $\hat{y}_{\text{cal}}$

---

## E   IMPLEMENTATION DETAILS

This appendix provides further details on the datasets, baseline methods, and experimental configurations.

**Dataset descriptions**

- BNCI2014001: Derived from BCI Competition IV-2a Tangermann et al. (2012), this dataset includes data from 9 subjects performing four-class MI tasks (left hand, right hand, feet, tongue) recorded using 22 EEG electrodes at 250 Hz. Each subject completed two sessions (288 trials per session) with a trial duration of 4 seconds. Only trials of left hand and right hand were used for analysis.

- BNCI2014002: This dataset contains data from 14 subjects performing two-class MI tasks (right hand vs. feet) recorded using 15 EEG electrodes at 512 Hz. Each subject participated in one session comprising 160 trials, with a trial duration of 5 seconds. Only the first 100 trials per subject were used for analysis.

- BNCI2015001: This dataset comprises data from 12 subjects performing two-class MI tasks (right hand vs. feet) recorded using 13 EEG electrodes at 512 Hz, each subject completed three sessions with 400 trials per session and a trial duration of 5 seconds.

- SHU MI Dataset (Ma et al., 2022). This dataset is particularly relevant as it was specifically designed to study long-term variability, containing data from 25 subjects recorded over 5 different days. Following the evaluation protocol provided with the dataset's official code, we performed a rigorous Leave-One-Subject-Out (LOSO) experiment on the last 10 subjects.

**Baselines**   To comprehensively evaluate BTTA-DG, we compared it against:

- CSP Blankertz et al. (2007): A traditional machine learning method that constructs spatial filters to maximize the variance of one class while minimizing that of another, thereby enhancing class separability.

- EEGNet Lawhern et al. (2018): A compact convolutional neural network that integrates spatial-temporal feature extraction from EEG signals. BN-adapt Schneider et al. (2020): Adjusts BN statistics to better adapt the target distribution.

- EEG Conformer (Song et al., 2022): A Transformer-based architecture designed to capture complex temporal dependencies in EEG signals. This baseline tests whether architectural advances alone are sufficient to overcome the domain shift challenge.

- Tent Wang et al. (2020): Optimizes affine parameters of BN via entropy minimization.

- PL Lee et al. (2013): Uses high-confidence predictions as pseudo-labels for self-training.

- SAR Niu et al. (2023): Incorporates reliable entropy minimization and sharpness-aware optimization to suppress noisy samples with large gradients.

- CoTTA Wang et al. (2022): Adapts the model using pseudo-labels generated through data augmentation and stochastic weight restoration to mitigate catastrophic forgetting.
  Data augmentation techniques implemented include Gaussian noise, time shift, frequency shift, phase perturbations, time dropout, channel dropout, channel shuffle, bandstop filter, and channel symmetry Rommel et al. (2022).

- T-TIME Li et al. (2023): Uses ensemble learning where multiple classifiers predict each unlabeled test EEG sample, updating classifiers via conditional entropy minimization and adaptive marginal distribution regularization.

- OTTA Wimpff et al. (2024): Integrats Euclidean Alignment or Riemannian Alignment techniques with entropy minimization of BN finetuning to reduce cross-subject domain shifts.

**Pre-training and test-time adaptation configurations**   The SincAdaptNet was pre-trained on train set using the AdamW optimizer with a learning rate of 1e-3, a batch size of 32, for 100 epochs. For fair comparison, all TTA methods (except CSP and EEGNet) used SincAdaptNet as the backbone and applied EA preprocessing. SincAdaptNet processes EEG trials $s \in \mathbb{R}^{B \times 1 \times C \times T}$ through:

(1) Spatial filtering via 2D convolution (kernel_size $= (C, 1)$, $F_{\text{spat}} = 16$ outputs), (2) Temporal filtering with Sinc FIR bandpass filters (4-30Hz) via SincConv (kernel_size $= (1, 51)$, $F_{\text{sinc}} = 16$ outputs), (3) Layer normalization, (4) Increasing-Channel convolution with ELU activation, (5) Layer normalization, and (6) Classifing convolution with LogSoftmax activation.

During test-time adaptation, the model was adapted using a batch size of 1, for 1 epoch. Compared with mini-batch settings, this single-trial setting is both more realistic and more challenging, because many TTA methods (e.g., batch-norm–based approaches like BN-adapt) rely on batch statistics and degrade or fail when only one sample is available. In contrast, our method works robustly with batch size of 1 thanks to the use of layer normalization and a probabilistic GMM over Dirichlet features instead of batch statistics. The same adaptation mechanism can be directly applied to larger mini-batches without any modification.. The memory bank size was set to 1000. The convergence tolerance for Dirichlet parameter estimation was set to $\varepsilon = 1e - 3$. As shown in the sensitivity analysis in Figure 4, our method is robust to moderate variations in hyperparameters of our adaptation process. The final values for GMM components, confidence threshold, and entropy threshold were determined based on a grid search over a reasonable range, with the optimal values selected based on the average performance on a held-out validation set created from the source domain subjects. This standard validation procedure ensures that the hyperparameter tuning is fair and does not use any information from the target domain. The GMM was configured with 8 components. The minimum confidence threshold $\tau_{\text{conf}}$ was set to 0.596, and the maximum entropy threshold $\tau_{\text{ent}}$ was set to 0.673.

Experiments were implemented via Python 3.10 and PyTorch 2.1.0, and ran on a server with NVIDIA TITAN V GPU and an Intel(R) Xeon(R) Gold 6230 CPU @ 2.10GHz.

## F    Tables of Cross-subject Adaptation Accuracy and Statistical Significance Analysis

Table 11 summarize LOSO cross-subject accuracies on SHU MI dataset for online adaptation methods. Our BTTA-DG framework consistently outperforms all competing TTA methods. However, the performance gain is not statistically significant. We have investigated this and believe it stems from the lower EEG quality and higher complexity (5 sessions) of this particular dataset. This has limited the effectiveness of the feature extractors. When the baseline feature discriminability is lower, it becomes inherently more difficult for any TTA method to demonstrate large, statistically significant gains.

We assess statistical significance by conducting pairwise one-way ANOVA tests between BTTA-DG and each baseline. Table 12 , 13 and 14 list per-subject and overall ANOVA $p$-values comparing BTTA-DG against each baseline. In most subjects, BTTA-DG demonstrated a significant difference from the baseline method ($p < 0.05$), further validating the statistical superiority of the BTTA-DG.

Table 11: Cross-subject adaptation accuracy (%) on SHU MI dataset.

| Setting | Method | Avg. (%) |
|---|---|---|
| Source | CSP | 59.24 |
| | EEGNet | $61.07_{\pm 1.10}$ |
| | EEG conformer | $61.36_{\pm 1.65}$ |
| | SincAdaptNet | $62.42_{\pm 1.72}$ |
| Online TTA | BN-adapt | $62.42_{\pm 1.72}$ |
| | Tent | $56.23_{\pm 2.96}$ |
| | PL | $56.65_{\pm 2.82}$ |
| | CoTTA | $62.73_{\pm 1.78}$ |
| | SAR | $58.33_{\pm 2.37}$ |
| | T-TIME | $62.37_{\pm 1.75}$ |
| | OTTA | $63.29_{\pm 1.20}$ |
| | **BTTA-DG** | $\mathbf{64.06_{\pm 1.92}}$ |

Table 12: Pairwise ANOVA $p$-values comparing BTTA-DG to each baseline on BNCI2014001 for cross-subject adaptation.

| Method | S0 | S1 | S2 | S3 | S4 | S5 | S6 | S7 | S8 | Overall |
|---|---|---|---|---|---|---|---|---|---|---|
| SincAdaptNet | 3.5e-07 | 1.2e-08 | 8.4e-11 | 9.0e-01 | 6.2e-02 | 5.1e-02 | 2.2e-02 | 1.2e-04 | 1.7e-04 | 1.1e-03 |
| BN-adapt | 3.5e-07 | 1.2e-08 | 8.4e-11 | 9.0e-01 | 6.2e-02 | 5.1e-02 | 2.2e-02 | 1.2e-04 | 1.7e-04 | 1.1e-03 |
| Tent | 1.4e-06 | 6.1e-08 | 6.8e-04 | 1.6e-10 | 2.5e-07 | 6.9e-07 | 9.7e-12 | 7.6e-05 | 2.0e-08 | 4.6e-05 |
| PL | 1.5e-07 | 7.4e-11 | 2.7e-06 | 8.5e-07 | 4.6e-04 | 2.1e-06 | 1.6e-06 | 3.7e-03 | 3.9e-05 | 4.9e-04 |
| CoTTA | 9.5e-07 | 2.9e-05 | 1.1e-05 | 1.9e-01 | 5.7e-01 | 1.6e-03 | 8.3e-02 | 8.0e-03 | 2.9e-03 | 1.2e-03 |
| SAR | 9.9e-06 | 7.3e-05 | 9.0e-05 | 4.4e-02 | 3.1e-07 | 7.6e-05 | 7.2e-02 | 6.6e-03 | 3.0e-02 | 2.7e-04 |
| T-TIME | 1.1e-05 | 4.1e-05 | 4.4e-04 | 1.3e-01 | 2.6e-01 | 2.0e-07 | 1.3e-07 | 4.1e-03 | 1.4e-03 | 1.5e-03 |
| OTTA | 2.3e-06 | 4.7e-07 | 1.7e-08 | 3.8e-01 | 9.9e-01 | 1.4e-01 | 1.1e-03 | 2.4e-04 | 4.1e-02 | 4.8e-02 |

Table 13: Pairwise ANOVA $p$-values comparing BTTA-DG to each baseline on BNCI2014002 for cross-subject adaptation.

| Method | S0 | S1 | S2 | S3 | S4 | S5 | S6 | S7 | S8 | S9 | S10 | S11 | S12 | S13 | Overall |
|---|---|---|---|---|---|---|---|---|---|---|---|---|---|---|---|
| SincAdaptNet | 4.5e-02 | 2.9e-10 | 0.00 | 5.8e-05 | 1.6e-04 | 3.5e-07 | 3.7e-09 | 1.0e-07 | 0.00 | 1.7e-04 | 9.0e-08 | 5.8e-04 | 9.4e-05 | 3.2e-07 | 4.4e-07 |
| BN-adapt | 4.5e-02 | 2.9e-10 | 0.00 | 5.8e-05 | 1.6e-04 | 3.5e-07 | 3.7e-09 | 1.0e-07 | 0.00 | 1.7e-04 | 9.0e-08 | 5.8e-04 | 9.4e-05 | 3.2e-07 | 4.4e-07 |
| Tent | 8.2e-11 | 1.8e-08 | 1.3e-04 | 8.9e-07 | 9.3e-08 | 1.3e-03 | 6.7e-04 | 2.0e-08 | 1.2e-02 | 6.0e-08 | 9.7e-08 | 6.0e-06 | 6.4e-13 | 1.9e-13 | 2.3e-07 |
| PL | 1.3e-09 | 6.3e-07 | 3.3e-01 | 1.6e-06 | 1.9e-06 | 5.8e-04 | 1.1e-05 | 1.5e-08 | 9.0e-08 | 6.7e-07 | 2.5e-03 | 2.0e-09 | 1.9e-13 | 1.8e-05 |
| CoTTA | 3.8e-03 | 1.4e-05 | 3.3e-01 | 1.3e-02 | 5.6e-03 | 1.4e-03 | 1.1e-05 | 3.9e-05 | 1.0e+00 | 1.5e-02 | 9.9e-04 | 1.2e-03 | 8.0e-04 | 5.4e-03 | 4.1e-05 |
| SAR | 2.2e-03 | 3.1e-08 | 8.7e-02 | 5.4e-04 | 1.2e-02 | 1.9e-03 | 2.2e-05 | 2.6e-06 | 3.7e-02 | 2.0e-04 | 2.2e-07 | 3.1e-06 | 1.9e-06 | 4.0e-04 | 7.2e-06 |
| T-TIME | 5.6e-03 | 9.4e-05 | 3.7e-02 | 2.6e-02 | 7.3e-01 | 1.4e-04 | 6.8e-06 | 2.4e-04 | 5.9e-08 | 3.9e-06 | 2.6e-06 | 8.0e-04 | 6.4e-06 | 2.4e-05 | 5.9e-06 |
| OTTA | 2.0e-02 | 1.0e+00 | 2.7e-06 | 2.6e-04 | 8.8e-03 | 4.8e-01 | 1.5e-01 | 6.9e-08 | 6.5e-02 | 4.2e-04 | 2.1e-11 | 3.3e-01 | 1.2e-01 | 2.3e-05 | 3.7e-02 |

Table 14: Pairwise ANOVA $p$-values comparing BTTA-DG to each baseline on BNCI2015001 for cross-subject adaptation.

| Method | S0 | S1 | S2 | S3 | S4 | S5 | S6 | S7 | S8 | S9 | S10 | S11 | Overall |
|---|---|---|---|---|---|---|---|---|---|---|---|---|---|
| SincAdaptNet | 5.8e-08 | 4.6e-07 | 2.9e-02 | 3.8e-08 | 8.3e-02 | 6.2e-03 | 3.1e-02 | 4.8e-03 | 3.2e-01 | 2.9e-05 | 2.8e-05 | 7.4e-11 | 5.5e-04 |
| BN-adapt | 5.8e-08 | 4.6e-07 | 2.9e-02 | 3.8e-08 | 8.3e-02 | 6.2e-03 | 3.1e-02 | 4.8e-03 | 3.2e-01 | 2.9e-05 | 2.8e-05 | 7.4e-11 | 5.5e-04 |
| Tent | 8.2e-06 | 1.5e-10 | 2.2e-06 | 6.0e-14 | 1.1e-09 | 7.7e-10 | 4.4e-10 | 7.0e-11 | 5.5e-09 | 1.3e-08 | 8.4e-18 | 2.0e-21 | 7.0e-07 |
| PL | 1.3e-02 | 2.7e-02 | 2.7e-05 | 8.8e-04 | 1.6e-12 | 1.4e-07 | 4.2e-06 | 1.8e-11 | 9.8e-11 | 8.5e-08 | 3.0e-17 | 4.7e-22 | 8.0e-06 |
| CoTTA | 9.9e-03 | 3.7e-02 | 1.9e-01 | 1.3e-04 | 5.7e-01 | 1.6e-02 | 2.8e-02 | 2.9e-02 | 7.6e-01 | 2.5e-02 | 1.1e-07 | 1.8e-09 | 4.6e-03 |
| SAR | 1.2e-02 | 6.2e-03 | 1.5e-02 | 3.9e-06 | 6.6e-02 | 3.6e-01 | 1.1e-02 | 5.0e-03 | 5.7e-01 | 7.3e-04 | 2.0e-03 | 4.3e-07 | 1.2e-02 |
| T-TIME | 1.7e-06 | 7.0e-05 | 2.9e-03 | 5.4e-07 | 2.9e-07 | 1.2e-04 | 1.4e-01 | 4.2e-05 | 7.0e-03 | 5.5e-07 | 2.8e-03 | 2.7e-07 | 3.1e-05 |
| OTTA | 1.9e-02 | 3.9e-07 | 2.1e-03 | 1.9e-06 | 4.3e-02 | 5.5e-14 | 1.3e-06 | 8.9e-01 | 2.2e-07 | 4.4e-02 | 2.2e-03 | 2.8e-12 | 5.0e-02 |

# G ADAPTATION ON LOW-CHANNEL EEG

Given the practical constraints of portable EEG hardware, we further evaluated BTTA-DG on BCI Competition IV 2b, which contains only three EEG channels. We followed the same LOSO cross-subject protocol as in the main experiments. All architectural, training, and adaptation hyperparameters were kept identical, except for adjusting the input channel dimension.

Table 15 reports cross-subject accuracies for source models and online TTA baselines. Despite the extremely low sensor density, BTTA-DG achieves the highest average accuracy of 78.76%, with statistically significant gains over the SincAdaptNet source model. These results confirm that the proposed Dirichlet–GMM adaptation mechanism remains effective even under minimal channel configurations, supporting its practical relevance for low-density BCI systems.

Table 15: Cross-subject adaptation accuracy (%) on BCI Competition IV 2b (3 channels). An asterisk (*) denotes statistical significance leverl (*: $p < 0.05$).

| Setting | Method | S0 | S1 | S2 | S3 | S4 | S5 | S6 | S7 | S8 | Avg. |
|---|---|---|---|---|---|---|---|---|---|---|---|
| Source | CSP | 70.09 | 61.25 | 63.68 | 77.17 | 78.21 | 68.21 | 74.20 | 73.82 | 72.51 | 71.01 |
| | EEGNet | 74.28 | 63.90 | 68.32 | 82.35 | 80.16 | 70.05 | 77.84 | 76.61 | 75.10 | $74.29_{\pm 1.46}$ |
| | EEG conformer | 72.23 | 62.45 | 66.17 | 83.12 | 79.21 | 69.10 | 76.30 | 76.56 | 76.56 | $73.61_{\pm 2.62}$ |
| | SincAdaptNet | 74.57 | 64.73 | 72.07 | 89.67 | 80.93 | 71.32 | 79.72 | 80.00 | 73.85 | $76.32_{\pm 0.89}$ |
| Online TTA | BN-adapt | 74.57 | 64.73 | 72.07 | 89.67 | 80.93 | 71.32 | 79.72 | 80.00 | 73.85 | $76.32_{\pm 0.89}$ |
| | Tent | 67.27 | 65.22 | 66.60 | 69.75 | 67.42 | 66.18 | 68.27 | 71.58 | 67.85 | $67.79_{\pm 1.24}$ |
| | PL | 67.77 | 65.48 | 67.08 | 70.52 | 70.33 | 66.57 | 73.17 | 72.70 | 68.45 | $69.12_{\pm 1.63}$ |
| | CoTTA | 74.78 | 64.73 | 72.30 | 90.05 | 82.02 | 71.93 | 80.38 | 80.30 | 74.65 | $76.79_{\pm 1.29}$ |
| | SAR | 75.08 | 67.42 | 71.88 | 79.02 | 81.03 | 74.03 | 80.48 | 76.83 | 69.30 | $75.01_{\pm 3.01}$ |
| | T-TIME | 75.92 | 64.18 | 72.00 | **91.68** | 80.53 | 72.85 | 81.02 | 79.78 | 72.92 | $76.77_{\pm 1.53}$ |
| | OTTA | **78.55** | **67.60** | 70.78 | 84.78 | 81.42 | 67.70 | 81.98 | 79.00 | 75.32 | $76.34_{\pm 1.05}$ |
| | BTTA-DG | 77.50 | 67.00 | **74.33**$^*$ | 91.50 | **84.17**$^*$ | **76.50**$^*$ | 83.58 | **82.83**$^*$ | 76.50 | $\mathbf{78.76}^*_{\pm 1.69}$ |

## H WITHIN-SUBJECT CROSS-SESSION EXPERIMENTS

To verify that BTTA-DG can adapt not only across subjects but also across sessions for the same subject, we conducted a within-subject cross-session study on BNCI2014001. Specifically, we pretrained SincAdaptNet on 1st session of each subject and then performed test-time adaptation on that subject's 2nd session, simulating session shifts on different days. Table 5 reports the classification accuracies for cross-session adaptation. BTTA-DG achieves an average of $86.50\% \pm 2.49\%$, significantly outperforming all competing TTA methods (asterisks denote $p < 0.05$ by pairwise ANOVA against each baseline).

Table 16 presents the corresponding pairwise ANOVA $p$-values comparing BTTA-DG to each baseline across subjects S0–S8, as well as an overall $p$-value computed on per-subject mean accuracies. Most comparison reach statistical significance, confirming that our gradient-free Bayesian calibration maintains its advantage even under within-subject cross-session drift.

Table 16: Pairwise ANOVA $p$-values comparing BTTA-DG to each baseline on BNCI2014001 for cross-session adaptation.

| Method | S0 | S1 | S2 | S3 | S4 | S5 | S6 | S7 | S8 | Overall |
|---|---|---|---|---|---|---|---|---|---|---|
| SincAdaptNet | 1.7e-01 | 3.2e-07 | 8.6e-09 | 3.1e-04 | 1.6e-03 | 2.2e-06 | 6.2e-04 | 1.1e-11 | 1.9e-06 | 1.8e-03 |
| BN-adapt | 1.7e-01 | 3.2e-07 | 8.6e-09 | 3.1e-04 | 1.6e-03 | 2.2e-06 | 6.2e-04 | 1.1e-11 | 1.9e-06 | 1.8e-03 |
| Tent | 1.6e-01 | 1.0e-14 | 6.5e-04 | 1.3e-11 | 1.1e-15 | 2.4e-11 | 1.1e-10 | 3.6e-02 | 8.7e-03 | 4.5e-03 |
| PL | 3.2e-06 | 3.8e-14 | 1.0e-06 | 4.2e-12 | 4.7e-05 | 1.7e-07 | 1.9e-05 | 1.1e-03 | 5.6e-02 | 3.0e-03 |
| CoTTA | 7.8e-01 | 9.8e-09 | 8.4e-04 | 3.8e-05 | 3.4e-03 | 9.4e-06 | 1.6e-03 | 2.5e-09 | 3.5e-03 | 7.2e-03 |
| SAR | 2.8e-01 | 1.6e-05 | 8.9e-04 | 8.3e-04 | 9.6e-05 | 3.2e-06 | 3.1e-04 | 1.3e-07 | 7.6e-04 | 3.7e-03 |
| T-TIME | 9.7e-06 | 1.5e-08 | 2.5e-05 | 5.6e-04 | 2.9e-07 | 9.6e-06 | 3.0e-04 | 5.4e-07 | 9.6e-03 | 4.8e-04 |
| OTTA | 3.1e-10 | 3.0e-08 | 6.9e-17 | 3.3e-02 | 8.4e-01 | 4.0e-06 | 1.1e-01 | 3.0e-13 | 3.0e-09 | 3.7e-02 |

## I SLIDING-WINDOW ONLINE ADAPTATION

To directly address the concern about whole-trial processing versus low-latency online BCIs, we evaluated BTTA-DG in a sliding-window setting on BNCI2014001 under the same cross-subject setting as in the main experiments. Each 4 s trial was segmented into non-overlapping 1 s windows, and adaptation and inference were performed per window, while all other training and adaptation settings were kept identical.

Table 17 reports window-level accuracies for the source models and online TTA baselines. As expected, overall accuracies are lower than in the full-trial setting. Nevertheless, BTTA-DG still achieves the best average performance (76.49%) and provides a clear improvement over the SincAdaptNet source model (74.52%). This shows that the proposed Dirichlet–GMM adaptation mechanism is compatible with online windows (batch size = 1) and can effectively operate in a low-latency, continuous-feedback regime, mitigating the practical limitation that the main experiments use full trials.

Table 17: Sliding-window adaptation accuracy (%) on BNCI2014001 with 1-s windows. An asterisk (*) denotes statistical significance level (*: $p < 0.05$).

| Setting | Approach | S0 | S1 | S2 | S3 | S4 | S5 | S6 | S7 | S8 | Avg. |
|---|---|---|---|---|---|---|---|---|---|---|---|
| Source | CSP | 74.17 | 66.48 | 79.02 | 61.25 | 55.44 | 74.44 | 67.99 | 83.78 | 83.95 | 71.84 |
| | EEGNet | 74.80 | 67.13 | 80.31 | 61.74 | 57.60 | 75.64 | 69.90 | 85.95 | 85.33 | $73.16 \pm 1.35$ |
| | EEGConformer | 75.80 | 67.63 | 80.91 | 62.04 | 57.80 | 75.94 | 70.40 | 86.55 | 86.03 | $73.68 \pm 1.94$ |
| | SincAdaptNet | 76.60 | 68.33 | 81.81 | 62.64 | 58.40 | 76.74 | 70.90 | 87.85 | 87.43 | $74.52 \pm 1.30$ |
| Online TTA | BN-adapt | 76.60 | 68.33 | 81.81 | 62.64 | 58.40 | 76.74 | 70.90 | 87.85 | 87.43 | $74.52 \pm 1.30$ |
| | Tent | 70.97 | 63.61 | 76.25 | 57.71 | 54.72 | 71.60 | 64.87 | 79.44 | 79.51 | $68.74 \pm 2.89$ |
| | PL | 73.40 | 63.89 | 78.47 | 58.33 | 54.10 | 72.29 | 66.18 | 81.25 | 85.00 | $70.32 \pm 3.26$ |
| | CoTTA | 76.83 | 68.85 | 82.18 | 63.08 | 57.94 | 76.97 | 70.93 | 88.43 | 88.22 | $74.82 \pm 1.50$ |
| | SAR | 76.86 | 66.24 | 79.01 | 60.54 | 56.86 | 74.22 | 70.61 | 83.39 | 84.01 | $72.42 \pm 2.29$ |
| | T-TIME | 76.00 | 67.53 | 83.01 | 62.60 | 57.04 | 76.97 | 69.61 | 89.89 | 87.73 | $74.48 \pm 1.68$ |
| | OTTA | 76.63 | 67.42 | 81.58 | 63.39 | 52.00 | 75.33 | 70.06 | 89.68 | **89.96** | $74.01 \pm 1.96$ |
| | BTTA-DG | **77.39** | **72.53*** | **83.64** | **64.89** | **60.03** | **78.08** | **73.22*** | **90.58** | 87.81 | **76.49*** $\pm 1.41$ |

## J  WITHIN-SUBJECT k-FOLD ADAPTATION ON BNCI2014001

To understand the intrinsic "decodeability" of each subject and its relationship with TTA effectiveness, we conducted a three-fold within-subject cross-validation on BNCI2014001. For each subject, we merged the two sessions, split trials into three folds, and performed within-subject training and testing. This setting enables a direct comparison with the cross-subject TTA results in the main text.

Table 18 summarizes within-subject accuracies across source models and online TTA baselines. Overall, within-subject training yields high decodeability for most subjects (e.g., SincAdaptNet reaches an average of 84.46%), while BTTA-DG further improves performance to 87.30%. For some subjects (e.g., S1, S2 and S3), the cross-subject SincAdaptNet accuracy in Table 2 exceeds their within-subject baselines. This observation is consistent with the hypothesis that pretraining on other subjects can benefit "bad-performing" subjects. For the majority of subjects, however, within-subject accuracies remain higher than cross-subject baselines, indicating that inter-subject variability can induce negative transfer and highlighting that the gains from BTTA-DG primarily stem from effective test-time adaptation.

Table 18: Within-subject k-fold adaptation accuracy (%) on BNCI2014001. An asterisk (*) denotes statistical significance over the SincAdaptNet source model (paired $t$-test, $p < 0.05$).

| Setting | Method | S0 | S1 | S2 | S3 | S4 | S5 | S6 | S7 | S8 | Avg. |
|---|---|---|---|---|---|---|---|---|---|---|---|
| Source | CSP | 90.26 | 51.53 | 95.81 | 67.48 | 70.65 | 74.23 | 72.80 | 93.55 | 88.41 | 78.30 |
| | EEGNet | 91.58 | 53.20 | 95.24 | 72.88 | 85.46 | 74.50 | 83.20 | 96.48 | 89.15 | $82.41_{\pm2.55}$ |
| | EEGConformer | 92.19 | 52.81 | 96.14 | 73.56 | 86.27 | 75.83 | 84.51 | 97.24 | 89.80 | $83.15_{\pm3.10}$ |
| | SincAdaptNet | 93.38 | 54.66 | 96.64 | 74.66 | 88.58 | 76.92 | 86.81 | 97.82 | 90.67 | $84.46_{\pm2.98}$ |
| Online TTA | BN-adapt | 93.38 | 54.66 | 96.64 | 74.66 | 88.58 | 76.92 | 86.81 | 97.82 | 90.67 | $84.46_{\pm2.98}$ |
| | Tent | 94.03 | 52.60 | 97.01 | 68.89 | 86.94 | 72.47 | 79.62 | 98.13 | 89.69 | $82.15_{\pm3.04}$ |
| | PL | 93.37 | 51.94 | 97.57 | 67.70 | 89.76 | 73.26 | 83.65 | 98.58 | 90.80 | $82.96_{\pm3.39}$ |
| | CoTTA | 94.20 | 54.82 | 97.60 | 74.90 | 90.14 | 77.81 | 87.81 | 98.65 | 91.35 | $85.25_{\pm2.85}$ |
| | SAR | 94.13 | 53.75 | 97.05 | 72.99 | 87.12 | 75.83 | 86.39 | 98.13 | 89.72 | $83.90_{\pm3.17}$ |
| | T-TIME | 92.53 | 52.60 | 97.29 | 72.19 | 89.65 | 76.42 | 85.69 | 98.68 | 89.90 | $83.89_{\pm3.40}$ |
| | OTTA | 93.00 | 57.84 | 97.38 | 77.18 | 92.28 | 76.41 | 91.86 | 98.08 | 90.68 | $86.08_{\pm2.35}$ |
| | BTTA-DG | 96.49* | 58.64 | 98.57 | 78.78* | 90.93 | 79.82* | 90.24 | 99.96 | 92.32 | $87.30_{\pm3.59}$ |

## K  FIGURES OF FREQUENCY RESPONSES OF THE LEARNED SINCCONV FILTERS

To provide a comprehensive view of how SincAdaptNet's learnable band-pass filters operate across all subjects, this appendix presents the complete frequency responses of each of the 24 Sinc-Conv filters for every LOSO test fold. These figures substantiate the interpretability and effectiveness of Sinc-Conv as a data-driven substitute for hand-crafted bandpass filtering in EEG analysis.

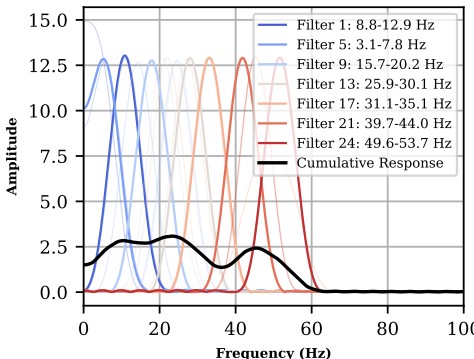
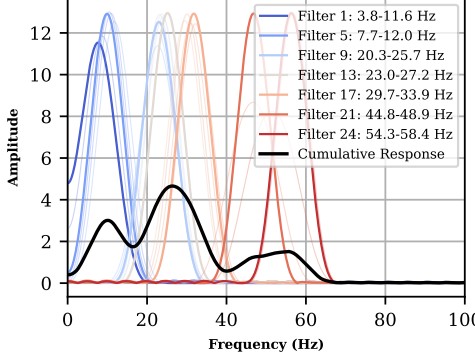

Figure 6: Training set: others; test set: S1          Figure 7: Training set: others; test set: S2

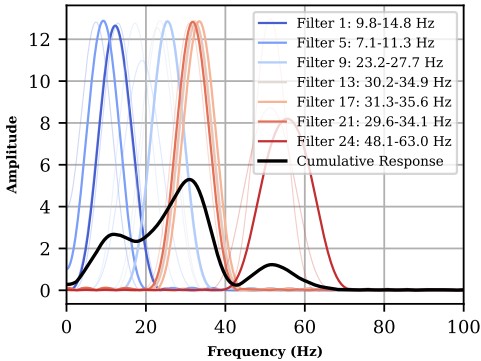

Figure 8: Training set: others; test set: S3

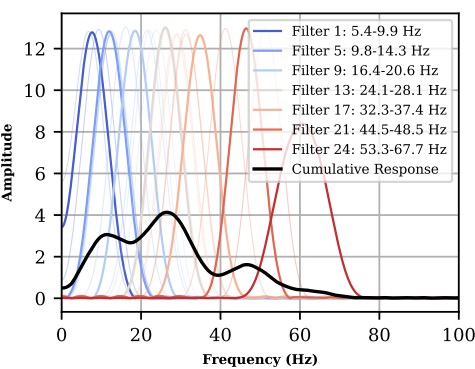

Figure 9: Training set: others; test set: S4

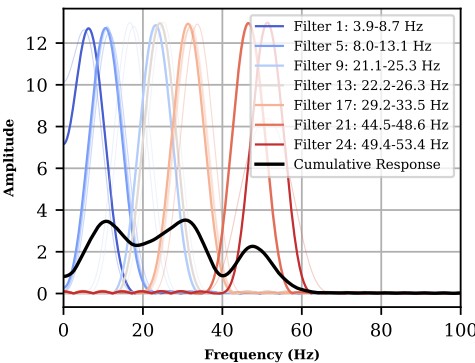

Figure 10: Training set: others; test set: S5

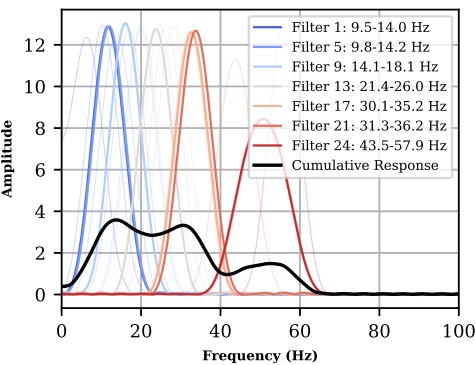

Figure 11: Training set: others; test set: S6

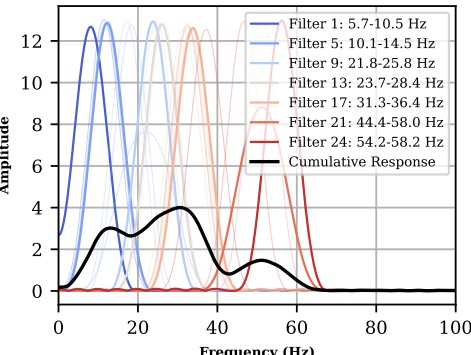

Figure 12: Training set: others; test set: S7

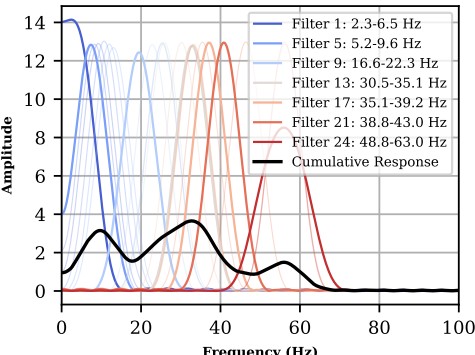

Figure 13: Training set: others; test set: S8

## L    QUANTITATIVE ALIGNMENT OF SINC FILTER PASSBANDS WITH MI-EEG RHYTHMS

To complement the main-text analysis in Table 6, we briefly describe how the quantitative alignment was computed. For each learned Sinc filter on BNCI2014001, we first obtained its lower and upper cut-off frequencies and then checked whether its passband overlapped a $\pm 2\,\text{Hz}$ tolerance around the standard $\mu$ (8–13 Hz), $\beta$ (13–30 Hz), and low-$\gamma$ (30–45 Hz) bands. For each subject, we then calculated the proportion of filters whose passbands fell into each of these bands or outside them, yielding the percentages reported in Table 6.

## M    TOPOGRAPHIC VISUALIZATION OF THE LEARNED SPATCONV KERNELS

The use of spatial convolution is a standard and effective technique in EEG deep learning models (e.g., EEGNet (Lawhern et al., 2018), ATCNet (Altaheri et al., 2022), EEG conformer (Song et al., 2022), M-FANet (Qin et al., 2024), and etc.) to learn spatial filters. While EEG channel ordering is arbitrary, the "C x 1" convolution learns a weighted sum of all channels (not only local adjacent channels), effectively creating data-driven spatial filters that can capture relationships between physically distant but functionally related brain regions (e.g., bilateral motor cortices). Our topographic visualizations in Figure 14 show that the learned kernels are not limited to local neighbors but capture diverse and physiologically meaningful scalp-wide patterns, confirming its ability to learn non-local inter-channel relationships.

Figure 14 shows all 16 spatial kernels from SincAdaptNet's Spat-Conv layer, rendered on a standard 10–20 montage Nomenclature (1991) scalp map. Together, these topographies span frontal, central, parietal, and occipital cortices, demonstrating that SincAdaptNet automatically learns multiple physiologically meaningful spatial filters—much like CSP but in a fully data-driven, end-to-end fashion.

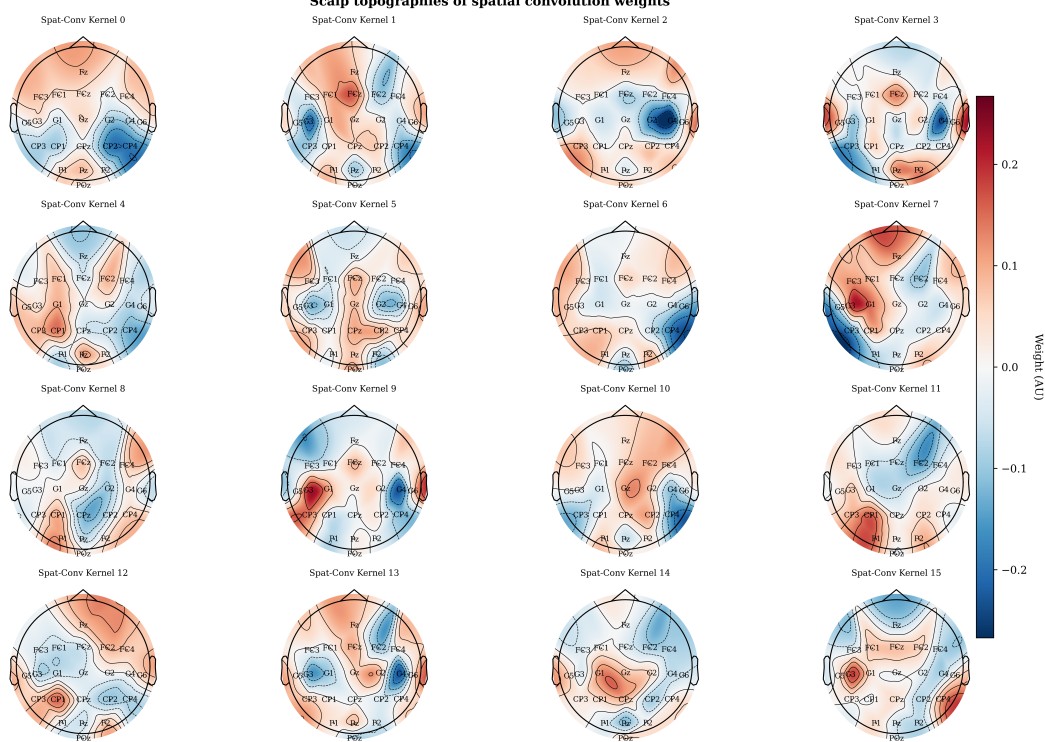

Figure 14: Topographic visualization of Spat-Conv kernels for SincAdaptNet trained on BNCI2014001 (22 channels). Each subplot displays one spatial kernel mapped onto the 2D electrode layout (standard 10-20 montage), where the color scale indicates the weight amplitude.

# N    FIGURES OF DIRICHLET PARAMETER DISTRIBUTION WITH GMM CLUSTERING

To provide a comprehensive view of how our Dirichlet–GMM pipeline captures class-specific structure in the low-dimensional embedding space across all subjects, this appendix presents the complete scatters of Dirichlet parameter representation and GMM clustering outcome for every LOSO test fold. These figures illustrate the stability and separability of the Dirichlet parameter representation across test trials, and how a Gaussian mixture model (GMM) clusters these parameters into coherent groups. This analysis validates our choice of Dirichlet projection to compactly encode prediction uncertainty and our use of GMM to preserve both global and local neighborhood information in BTTA-DG.

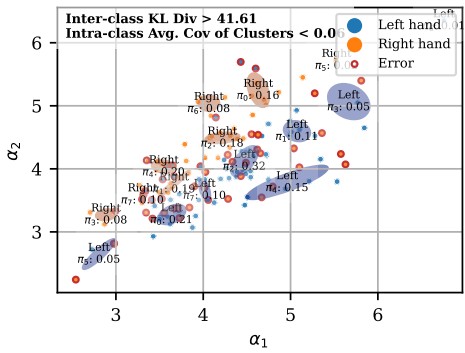

Figure 15: Training set: others; test set: S1

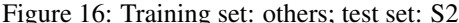

Figure 16: Training set: others; test set: S2

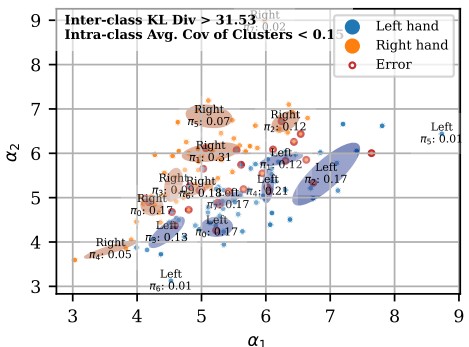

Figure 17: Training set: others; test set: S3

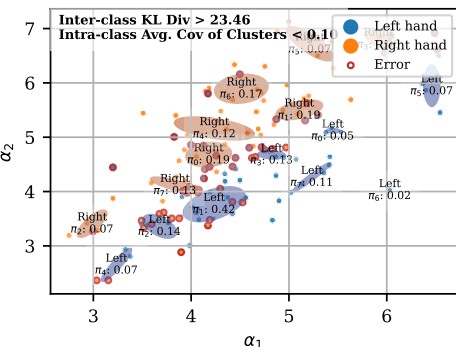

Figure 18: Training set: others; test set: S4

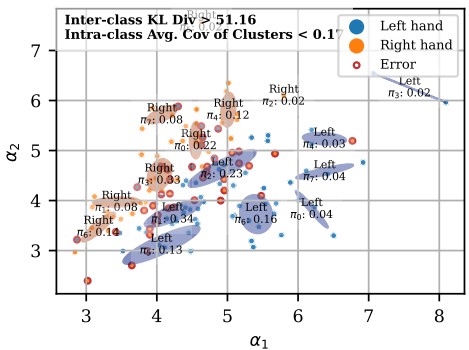

Figure 19: Training set: others; test set: S5

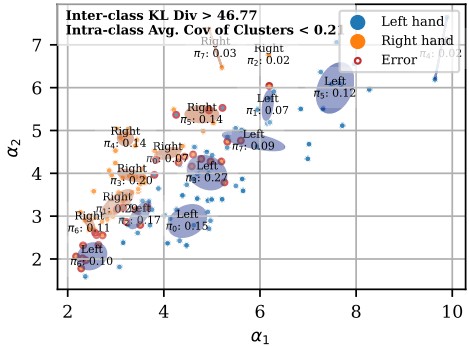

Figure 20: Training set: others; test set: S6

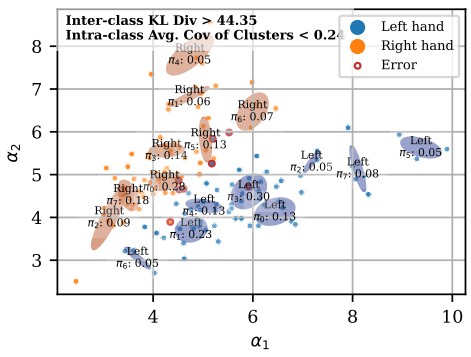

Figure 21: Training set: others; test set: S7

Figure 22: Training set: others; test set: S8

## O  COMPUTATIONAL COMPLEXITY

This appendix summarizes the space and time complexity of BTTA-DG's test-time pipeline, covering Dirichlet parameter estimation, memory maintenance, and GMM-based calibration. Let $M$ be the size of the memory bank, $|\mathcal{L}|$ the number of classes (i.e., the Dirichlet dimension $d = |\mathcal{L}|$), and $K$ the number of mixture components per class.

**Space complexity.**  For each trial we store only its low-dimensional Dirichlet parameter vector $\boldsymbol{\alpha} \in \mathbb{R}_+^d$, so the memory bank costs $\mathcal{O}(Md)$. GMM parameters per class add $\mathcal{O}(Kd)$ for means and $\mathcal{O}(Kd)$ for diagonal variances (plus $\mathcal{O}(K)$ for weights). Since $M$ is fixed and $d \in \{2, 4\}$ in our MI-EEG settings, the overall cost is minimal.

Concretely, in a 2-class MI task ($d = 2$) with a memory bank size of $M = 1000$ trials per class, the bank stores 2 buffers $\times$ 1000 trials $\times$ 2 floats/trial $\times$ 4 bytes/float32 values, corresponding to roughly 16 KB per subject. Even for $d = 4$ classes with the same $M$, the total memory remains on the order of 64 KB per subject. The GMM parameters with a small number of components $K$ add at most a few additional kilobytes. Since BTTA-DG is instantiated per target user at deployment, this memory cost scales linearly in the number of adapted users rather than in the size of the pre-training source user.

**Time complexity.**

- **Dirichlet MLE.** For a trial with $T$ time steps, computing sufficient statistics $s_i = \frac{1}{T} \sum_{j=1}^{T} \log x_{ij}$ costs $\mathcal{O}(Td)$; the fixed-point updates with Newton refinement for $\psi^{-1}$ then cost $\mathcal{O}(d\,I_{\text{iter}})$, where the number of iterations $I_{\text{iter}}$ is typically $< 5$–$10$ (see Appendix C). Hence, Dirichlet projection:  $\mathcal{O}(Td + d\,I_{\text{iter}}) \approx \mathcal{O}(Td)$.
- **Memory update.** Appending a single vector (and evicting the oldest when full) is a queue operation with $\mathcal{O}(1)$.
- **GMM re-fitting.** Re-fitting by EM over the memory bank has per-EM-iteration cost $\mathcal{O}(MKd^2)$. For $I_{\text{EM}}$ iterations, the per-trial overhead is $\mathcal{O}\big(I_{\text{EM}}\,MKd^2\big)$.

Given that $d \leq 4$, $K$ is small, and $M$ is bounded, the end-to-end overhead is negligible in practice while enabling robust, gradient-free adaptation. In terms of FLOPs, the Dirichlet projection and GMM steps contribute only a small fraction of the total cost relative to a SincAdaptNet forward pass, which is reflected in the modest increase from 133.3M to 141.6M FLOPs per trial reported for BN-adapt and BTTA-DG in Table 7.

## P  DETAILED ANALYSIS OF CLASS IMBALANCE SENSITIVITY

This appendix provides additional interpretation of the class-imbalance experiment reported in Table 9 in the main text.

As noted there, while the overall accuracy gracefully degrades with increasing imbalance, we observe that the accuracy for the minority class (Class 1) conversely improves as its prevalence decreases. For the **minority class**, our confidence and entropy thresholds ensure that only high-certainty trials of the rare class are added to its memory bank. This creates a highly "pure" and compact GMM, which becomes exceptionally good at identifying these specific, ideal minority-class trials.

Conversely, the GMM for the **majority class** must account for a much larger and more diverse set of trials, causing its distribution to become more diffuse. This can lead to a decrease in its own classification accuracy as the decision boundary shifts. Taken together, this finding highlights that our method does not simply fail under severe online class imbalance; instead, it adapts by specializing its model for rare events, which is a desirable robustness property for real-world applications.

## Q  DECLARATION OF LARGE LANGUAGE MODEL (LLM) USAGE

The LLM is used only for writing, editing, or formatting purposes and does not impact the core methodology, scientific rigorousness, or originality of the research.

