# OpenReview forum: "Bayesian Test-Time Adaptation via Dirichlet feature projection and GMM-Driven Inference for Motor Imagery EEG Decoding"
_ICLR.cc/2026/Conference — ICLR 2026 Poster_

### Official Review · Reviewer_G2CU · 2025-10-31

**Soundness:** 2
**Presentation:** 2
**Contribution:** 3
**Rating:** 4
**Confidence:** 4

**Summary:**

This paper proposes a novel method for cross-subject (and cross-session) Test Time Adaptation (TTA) of motor imagery BCIs, that is based on Bayesian-inference calibration of predictions, with the likelihood being given by the (GMM) distribution of test trials in the Dirichlet parameter space of their class probability distributions across time. This method is not gradient-based, avoiding high computational cost and catastrophic forgetting. Compared to other non-gradient-based methods in prior literature, authors claim that this approach can adequately capture high-dimensional (deep feature distribution) domain shifts. Results show state-of-the-art performance (+0.8%-2.6% absolute accuracy improvement) in cross-subject and cross-session accuracy compared to prior TTA methods, evaluated in an online setting across 4 public motor imagery datasets. Computational efficiency is the second best among all compared methods, and visualization of the GMM clusters of test trials in Dirichlet space shows good separation of classes. Finally, additional ablation studies are provided concerning the comparison of probability calibration vs. projection only and the impact of hyperparameters in estimating the GMM likelihood, thus explaining performance and design choices.

**Strengths:**

This paper offers substantial novelty to the field of TTA in BCIs, through creative combination and application of existing statistical modeling methods (Dirichlet distribution + GMM estimation + Bayesian inference) which advances the state of the art considerably on multiple datasets, while being interpretable and efficient at the same time.

The paper is of high quality; the methods proposed are well thought out and well explained, the experiments and figures provided support (most of) the claims well, and figures are both pleasant and informative (especially figure 1). Writing is very clear, convincing, and easy to follow, even when going through complex technical details.

**Weaknesses:**

1. **High-level statements misaligned with method details.** In various instances throughout the paper it is either implied (lines 20, 61, 63, 70, 74, 250) or directly stated (line 22) that the method “directly models the distribution of deep features”. However, this is not true from this reviewer’s understanding of the method - the Dirichlet parameter space, although reflecting the distribution of class probabilities across trial time (which is indeed more than just static trial predictions), is still only modeling low dimensional class probability features and not high-dimensional deep features of the trials as stated (or implied). If by “high-dimensional” the authors referred to the temporal dimension, that should be made more clear as the current phrasing can be misleading (usually when saying high-dimensional / deep features of a DNN this cannot be the output layer even if it makes predictions per token).
2. **DNN model choice not well motivated.** While the introduction states that “recent advances in large-scale EEG pretrained models [cited: labram, egg-gpt, etc.] trained on massive datasets have demonstrated unprecedented capabilities in learning general […] representations”, these models are not used as the backbone model or as baselines in the current paper’s experiments. Instead, a much shallower model is used (only 4 layers) and it is trained from scratch in the down-stream datasets. This choice should be better motivated (ideally also showing the performance of those large-scale pretrained models as baselines), and depending on what that motivation is, it could also be good to show TTA results using those models as backbones.
3. **Useful additional comparisons desired.** First, it would be useful to be able to compare the TTA performance with the performance achieved with full fine-tuning to the target subject, to quantify how much this gap is starting to close; for example EA+Fine-tuning achieves 82.6 in BNCI2014001 (see [1]). Additionally, it would be informative to know the (k-fold) within-subject performance of each of the subjects here (perhaps in the Appendix), to provide insights into the “decodeability” of each subject’s data and how it may relate to the effectiveness of TTA with this data. For example in [1], pretraining on the rest of the subjects was shown to especially benefit “bad-performing” subjects compared to their within-subject performance - here it could conversely be that those subjects are not very effective for TTA. Second, an additional ablation that performs the GMM-BTTA without the Dirichlet projection, using instead the feature space of the mean class-probability across trial time (red dot in Figure 1; e.g. (d) would be represented by (0, 0.5, 0) instead of (1, 5, 1)) would also be good to demonstrate the usefulness of the projection (uncertainty modeling) in the calibration.
4. **Presentation issues.** The conclusion paragraph needs some more work. Sentence “While the framework assumes” (line 481) is either left unfinished or has some grammatical error. Sentence “our framework demonstrates robustness under mild departures” (line 484) is not evidenced in the results (or is not made clear which result this refers to). Same line, “would” does not make sense. Figure 4 would benefit from having the scale of all y axes be the same so that the difference in fluctuations is more clear and comparable. The order of tables and figures in that page is also inconsistent with the flow of the text, should be Table 7, Table 6, Figure 4. The paper would benefit from having Table 13 in the main text. There should also be some comment in the text on why “BN-adapt” is that much faster (Table 7) than all others. On line 173 there is a mistake in the figure caption, “(b) and (d) illustrate” should be just “(d) illustrates”.

[1] Sartzetaki, C., Antoniadis, P., Antonopoulos, N., Gkinis, I., Krasoulis, A., Perdikis, S. and Pitsikalis, V., 2023. Beyond Within-Subject Performance: A Multi-Dataset Study of Fine-Tuning in the EEG Domain. In 2023 IEEE International Conference on Systems, Man, and Cybernetics (SMC) (pp. 4429-4435). IEEE.

**Questions:**

Please address the points outlined above in the “Weaknesses” section. If those are substantially addressed with a change in the paper’s content or with a convincing reason why this change is not feasible / desirable, the paper will be recommended for acceptance.

Especially addressing the 1st point is necessary for recommendation of acceptance. Depending on how the rest of them are addressed, the rating could be increased to either 6 or 8.

---

> ### Author Response · Authors · 2025-11-22
> **Response [1/5] to Official Review by Reviewer G2CU- Weakness 1**
>
> Dear Reviewer G2CU,
>
> We are sincerely grateful for your exceptionally thorough and insightful review. Your comments were not only constructive but also provided a clear roadmap that has enabled us to significantly strengthen the manuscript. The level of detail and clarity in your feedback is a testament to your expertise and diligent effort, for which we are particularly thankful. We have worked diligently to address each of your suggestions and believe the revised paper is substantially improved. We hope our corresponding revisions have addressed your concerns.
>
> **Addressing Weaknesses:**
>
> **1. High-level statements misaligned with method details.**
> Thank you for this this accurate and important comment. We fully acknowledge your comment and wish to clarify our original intention behind using the phrase “directly models the distribution of deep features.” Our aim was to highlight that BTTA-DG operates on the distribution of temporal predictive embeddings, which is conceptually different from methods that solely rely on data alignment, batch normalization statistics or static predictions. We intended to convey that our approach captures the evolution of these embeddings across time, providing a deeper and more dynamic view. We regret the confusion caused by our phrasing and respectfully request that you and the area chair consider this as a phrasing issue, which is not fundamental flaw in our method.
>
> The temporal predictive embeddings in our method are conceptually similar to typical deep features, with the key difference being the added constraint imposed by the softmax function. These “high-dimensional” embeddings are generated by applying a Cls-Conv convolutional layer followed by a softmax activation, operating on higher-dimensional hidden representations. Concretely, our adaptation operates on the temporal predictive embeddings $X \in \mathbb{R}^{|\mathcal{L}| \times T}$, where $T$ represents the temporal dimension (e.g., $250 \text{Hz} \times 4 \text{s} = 1000$ time samples). For a two-class problem, this results in a $2 \times 1000$ tensor per trial, which gives a 2000-dimensional feature (when flattened). This is not a low-dimensional feature overall. Additionally, the class dimension $|\mathcal{L}|$ can scale with the number of classes, further increasing the feature dimensionality. The Dirichlet parameters $\alpha$ then summarizes the distribution of temporal predictive embeddings rather than just static outputs. Therefore, we apologize for the misuse of high-dimensional deep features.
>
> We fully agree with your rigorous interpretation that referring to these as “deep features” might suggest operations directly on high-dimensional hidden “deep features”, which is not the case in our framework. **We have replaced the term “deep features” with “temporal predictive embeddings” in the revised manuscript, and we also remove “high-dimensional” in our context.** We hope this clarifies the distinction and maintains the core message that BTTA-DG models how predictive distributions evolve over time within each trial, rather than relying solely on data alignment, batch-level statistics, or static predictions.

---

> ### Author Response · Authors · 2025-11-22
> **Response [2/5] to Official Review by Reviewer G2CU- Weakness 2**
>
> **Addressing Weaknesses (continued):**
>
> **2. DNN model choice not well motivated.**
>
> **2a. Why we mentioned large-scale EEG pretrained models in the introduction but did not use them as baselines.**
> We appreciate your insightful comment regarding our reference to large-scale pretrained EEG models. The purpose of mentioning these models in the introduction was to situate our work within current trends in the field. While pretrained models show the impressive capabilities in learning general representation, their application to downstream BCI tasks is not yet a solved problem. In fact, our survey found that their performance on specific downstream tasks (like Motor Imagery) is not always significantly superior to a lightweight, task-specific model trained from scratch. To illustrate this, we have summarized reported results in **Table A** below. We believe this performance gap stems from the extremely high variability of EEG signals, which are fundamentally different from text or images.
>
> **Table A: Comparison of pretrained models vs. lightweight models on BCIC-IV-2a-4class and BCIC-2b (LOSO).**
>
> | Scale                          | Methods       | Number of Parameters | BCIC-IV-2a-4class | BCIC-2b |
> |-------------------------------|--------------|----------------------|-------------------|---------|
> | Large-scale foundation models | BENDR [a]     | 4 M                  | 0.4899            | 0.7067  |
> |   | BIOT [b]      | 3.2 M                | 0.4590            | 0.6409  |
> |   | LaBraM [c]    | 5.8 M                | 0.5613            | 0.6851  |
> |   | EEGPT [d]     | 25 M                 | 0.5846            | 0.7212  |
> |   | CBraMod [e,f] | 4.0 M                | 0.5585            | 0.6735  |
> | Lightweight models            | EEGNet [g]    | 0.003 M              | 0.5685            | 0.7429  |
> |              | Conformer [h] | 0.55 M               | 0.5341            | 0.7361  |
> |              | SincAdaptnet  | 0.0015 M             | 0.5764            | 0.7632  |
>
> Another important consideration is that the community already has a growing body of work exploring large-scale EEG backbones. The field of EEG research requires diversity, and each researcher contributes based on their available resources and specific research interests. Our work focuses on the Bayesian TTA framework rather than the backbone architecture. We mentioned the large pretrained models in the introduction to frame this exact problem: high variability is a major challenge. Our paper proposes to tackle this challenge through an efficient, gradient-free adaptation mechanism, rather than through architectural scale. This explains both why we referenced the LPM trend and why they were not the focus of our baseline comparison.
>
> If mentioning large-scale pretrained models (but not using them as baselines) still seems inappropriate, we are open to removing it, as we primarily aim to focus on TTA, which is the core contribution of our work.
>
> ---
> [a] Demetres Kostas, et al. BENDR: Using transformers and a contrastive self-supervised learning task to learn from massive amounts of eeg data. Frontiers in Human Neuroscience, 15:653659, 2021.
> [b] Chaoqi Yang, et al. BIOT: Biosignal transformer for cross-data learning in the wild. Advances in Neural Information Processing Systems, 36:78240–78260, 2023.
> [c] Wei-Bang Jiang, et al. Large brain model for learning generic representations with tremendous eeg data in bci. In The Twelfth International Conference on Learning Representations, 2024.
> [d] Guangyu Wang, et al. EEGPT: Pretrained transformer for universal and reliable representation of eeg signals. Advances in Neural Information Processing Systems, 37:39249–39280, 2024.
> [e] Jiquan Wang, et al. CBramod: A criss-cross brain foundation model for EEG decoding. In The Thirteenth International Conference on Learning Representations, 2025.
> [f] Ang Li, et al. Comet: A contrastive-masked brain foundation model for universal EEG representation. arXiv preprint arXiv:2509.00314, 2025.
> [g] Vernon J Lawhern, et al. EEGNet: a compact convolutional neural network for EEG-based brain–computer interfaces. Journal of neural engineering, 15(5):056013, 2018.
> [h] Song, Y., et al. EEG conformer: Convolutional transformer for EEG decoding and visualization. IEEE Transactions on Neural Systems and Rehabilitation Engineering, 31, 710-719.

---

> ### Author Response · Authors · 2025-11-22
> **Response [3/5] to Official Review by Reviewer G2CU- Weakness 2**
>
> **Addressing Weaknesses (continued):**
>
> **2b. Why a much shallower SincAdaptNet trained from scratch in the down-stream datasets.**
> We appreciate your point about motivating our use of a shallower SincAdaptNet trained from scratch on downstream datasets. This choice was driven by the need for a compact, efficient encoder tailored to online TTA of MI-EEG, where data are limited and variability is high, making very deep architectures prone to overfitting. From a statistical perspective, MI decoding is largely governed by low-dimensional structure in the time–frequency domain (μ/β rhythms over a small set of motor channels), so a very deep architecture is more likely to increase variance than to capture additional task-relevant structure under limited data. SincAdaptNet imposes a strong inductive bias by parameterizing its temporal filters with only two spectral parameters (low cut-off and bandwidth), favoring the band-limited solutions aligned with known MI neurophysiology. This constrains the hypothesis space, achieving a more favorable bias-variance trade-off.
>
> In addition, the benefits of frequency band extraction of SincAdaptNet for TTA are favorable. The learnable Sinc filters isolate the critical μ and β frequency bands, effectively filtering out irrelevant spectral noise in raw EEG signals. This noise reduction produces a more stable and less ambiguous sequence of predictive embeddings. A cleaner, more stable input to the accelerated fixed-point Iteration algorithm leads to a more robust estimate of the Dirichlet parameters, which are the features we adapt on. In other words, band-limited features improve the quality of the probabilistic representation on which our TTA operates, enhancing robustness to non-stationarity in realistic online BCI scenarios while keeping the overall system efficient and interpretable. **We have now made the above motivations explicit in _Appendix B (SincAdaptNet Details)_.**
>
> Empirically, this design does not weaken representation capabilities. Across all experimental settings, SincAdaptNet outperforms the Transformer-based EEG Conformer on five datasets in **Tables 2, 3, 4, 5, 10, 14, 16 and 17**. This demonstrates its ability to extract sufficiently discriminative features. Additionally, the lightweight architecture brings a practical benefit for online use. As shown in the expanded **Table 7**, SincAdaptNet achieves a full forward pass plus adaptation in about 15.7 ms per trial, which makes the overall system well suited for real time TTA scenarios.
>
> We agree that evaluating BTTA-DG on top of large pretrained EEG backbones would be interesting. However, doing so would require substantial additional computation for loading, adapting and tuning several large models across datasets, which is beyond what we could complete within the rebuttal period. Instead, we have reviewed results (e.g., **Table A**), showing pretrained accuracies comparable to ours. We note that prior TTA works such as T-TIME and OTTA also build on task specific lightweight backbones rather than large pretrained models. In future work, we commit to integrate BTTA-DG with large pretrained EEG models to explore whether the gains are additive.

---

> ### Author Response · Authors · 2025-11-22
> **Response [4/5] to Official Review by Reviewer G2CU- Weakness 3**
>
> **Addressing Weaknesses (continued):**
>
> **3. Useful additional comparisons desired.**
> Thank you for these constructive suggestions on additional comparisons, which helped us substantially strengthen the empirical analysis. We address each part of your comment below.
>
> **3a. Comparison with full fine-tuning on the target subject.**
> We agree that comparing TTA performance with full fine-tuning on the target subject is insightful, but such a comparison must respect the setting of test time adaptation, where the model has no access to source data, no pre-collected labeled calibration trials, and only receives unlabeled test trials sequentially. Our baselines already include gradient-based fine-tuning TTA methods implemented under exactly this setting (Tent, PL, CoTTA, SAR, T-TIME and OTTA), so their performance can be directly compared to BTTA-DG in realistic online scenarios. The approach in [i], which we cite in the revised **Appendix A Related Work**, provides a comprehensive multi-dataset study of fine-tuning and shows that even a small amount of labeled target data can substantially boost MI decoding performance, highlighting the potential of supervised adaptation when such labels are available. We see our unsupervised TTA framework as complementary to this line of work.
>
> **3b. Within-subject k-fold cross-validation.**
> We fully agree that conducting within-subject k-fold cross-validation is essential for understanding the intrinsic “decodeability” of each subject and its relationship with TTA effectiveness.  **We added a new appendix section (Appendix J) reporting within-subject three-fold cross-validation on the merged sessions of BNCI2014001.** The new **Table 17** there allows a direct comparison with the cross-subject TTA results in **Table 2**. We observe that the cross-subject SincAdaptNet accuracy for subjects S1 (63.93% vs. 54.66%), S2 (97.68% vs. 96.64%) and S3 (77.13% vs. 74.66%) exceeds their respective within-subject baselines. For these subjects, our findings are consistent with [i], where pretraining on other subjects particularly helps “bad-performing” subjects. However, most subjects still achieve higher accuracy in the within-subject setting, suggesting that cross-subject variability can induce negative transfer and that the additional gains in the cross-subject setting are mainly attributable to the effectiveness of BTTA-DG. Moreover, the above analysis from the perspectives of within-subject and cross-subject is somewhat indirect. As demonstrated in **Tables 2, 3, 4, 5, 10, 14, 16 and 17**, a more direct and controlled perspective on the effectiveness of TTA is provided by the comparison between SincAdaptNet and BTTA-DG, as these two baselines differ only in terms of the presence of the adaptation module.
>
> **3c. Ablation using the feature space of the mean class-probability across trial time.**
> Regarding the requested ablation that uses the mean class probability over time, we clarify that this corresponds exactly to our SincAdaptNet (source only) and SincAdaptNet+EA baselines, which operate purely on the time-averaged predictive distribution. These baselines consistently underperform the full BTTA-DG model across datasets. **In the revision, we have further strengthened this analysis in the expanded ablation study (Table 8).** We now report results on BNCI2014001 (cross-session and cross-subject), BNCI2014002, BNCI2015001 and SHU MI (cross-subject), and we add a BTTA-DG w/o EA variant that retains only the Dirichlet projection and GMM components. The updated **Table 8** shows that each component (EA, Dirichlet projection, GMM) contributes to the final gains, which demonstrates the usefulness of our BTTA-DG framework.
>
> ---
> [i] Christina Sartzetaki, et al. Beyond within-subject performance: A multi-dataset study of fine-tuning in the eeg domain. In 2023 IEEE International Conference on Systems, Man, and Cybernetics (SMC), pp. 4429–4435, 2023.

---

> ### Author Response · Authors · 2025-11-22
> **Response [5/5] to Official Review by Reviewer G2CU- Weakness 4**
>
> **Addressing Weaknesses (continued):**
>
> **4. Presentation issues.**
>
> **4a. Conclusion paragraph.**
> Thank you for highlighting these issues in the conclusion. We sincerely apologize for the grammatical error and the ambiguity regarding the claim of "robustness under mild departures." Our intention was to discuss the framework's reliance on balanced class distribution as a potential limitation. **We have rewritten this sentence in the revised Conclusion section to correct the grammar and ensure our claims are empirically supported.** Additionally, we replaced the awkward phrasing "would" with "could" to more accurately suggest future research directions.
>
> **4b. Figure 4 y-axes.**
> We agree that a consistent scale facilitates a better comparison of hyperparameter sensitivity. **We have unified the y-axis ranges across all subplots in _Figure 4_ in the revised manuscript** to make the fluctuations comparable.
>
> **4c. Ordering of tables and figures.**
> **We have reorganized the layout so that tables and figures now appear in the logical order referenced in the main text**, addressing the confusion in the original version.
>
> **4d. Moving Table 13 to the main text.**
> We agree that the class imbalance analysis provides critical insight into the robustness of our method. **We have moved the corresponding table (now Table 9 in the revised version; originally Table 13) from the Appendix to the Results section of the main text** to ensure it receives the appropriate attention.
>
> **4e. Why “BN-adapt” is much faster.**
> Thank you for pointing this out. **We have now clarified in the revised Computational efficiency paragraph that BN-adapt is the fastest method because it only recomputes batch-normalization statistics, without any gradient-based optimization or probabilistic inference**, which explains its low computational cost but also its relatively limited adaptation performance.
>
> **4f. Figure 1 Caption.**
> Thank you for catching this issue. Our intention was to emphasize that the contrast between the uniform distribution in (b) and the shifted distribution in (d) illustrates the concentration effect. To make this clear and to avoid the ambiguity, **we have revised the caption to: “Comparing (b) and (d) illustrates ...”**
>
> ---
>
> We again thank you for the considerable time and effort you devoted to reviewing our work. Your comments have led to substantial improvements in conceptual clarity, empirical validation, and presentation. We corrected the misaligned high-level statements, explained our backbone choice, added new comparisons, and cleaned up several presentation issues. We hope that our revisions address your concerns. We would be very grateful if you could consider these revisions when you do your final assessment. If any part of the response is unclear, we will clarify it for any further discussion at any time.

---

> > ### Comment · Reviewer_G2CU · 2025-11-27
> >
> > Thank you for your elaborate and detailed response.
> > My major concern regarding misalignment of claims about “deep feature” distribution modeling has been remedied with the changes in the revised manuscript, and it was indeed a phrasing issue rather than a fundamental methodological flaw. I believe that with the new phrasing, the contribution of the paper is more accurately outlined, and that this contribution is significant for the development of real-world BCIs, as well as methodologically novel and experimentally well-supported. Most of my other concerns have also been either remedied in the new version, or clarified in your response.
> >
> > Only one question and a couple of other more minor points remain.
> >
> > Q: About 3c, in my original question I referred to a setup which would perform GMM calibration on the mean class-probability across trial time instead of the Dirichlet projection (red dot in Figure 1; e.g. (d) would be represented by (0, 0.5, 0) instead of (1, 5, 1)). I believe this is not what SincAdaptNet (source only) does (there is no GMM calibration or test-time adaptation mechanism), could you clarify if my requested setup is for some reason logically impossible to implement?
> >
> > Point 1: I believe Table A of the response should be included in the appendix of the paper (perhaps in Related work?), as it very convincingly shows that SincAdaptNet is better than all large-scale models (apart from EEGPT with a less than 1% difference but x20k less parameters), which is not a given that every reader is aware of this.
> > This could also be discussed further in the Introduction - for example why *in MI specifically* large scale pretraining (counter to intuition) does not seem to help (yet) with handling variability.
> > Also this future work direction could be added in the Conclusion “In future work, we commit to integrate BTTA-DG with large pretrained EEG models to explore whether the gains are additive.”.
> >
> > Point 2: Line 73 fix grammatical coherence “temporal predictive embeddings shifts”
> >
> > **For now, I raise my score to 6, and I am open to a further raise awaiting response to the aforementioned points.**

---

> ### Author Response · Authors · 2025-11-29
> **Furthur Response [1/2] to Official Comment by Reviewer G2CU**
>
> Dear Reviewer G2CU,
>
> Thank you very much for your thoughtful comments. We are grateful that you regard the earlier issue about “deep feature” as a phrasing problem. We deeply appreciate your positive assessment that our work is significant for real-world BCIs and is both methodologically novel and experimentally well supported. We are encouraged that most of your concerns have been resolved by the revision and response. We are also sincerely thankful that you have raised your score to 6 and are open to a further increase, which really means a lot to us. Below we respond to your remaining question and points.
>
> **Q1. Ablation using mean class-probability with GMM calibration.**
> Thank you for this clarification. We apologize for having misunderstood your original question in our previous rebuttal. We had initially interpreted it as referring to direct classification using the mean class-probabilities without any test-time adaptation.
> **In the revised manuscript, we have added the requested variant, which we denote as “SincAdaptNet + EA + GMM”.** This model performs GMM-based Bayesian calibration directly on the mean class-probability across trial time (i.e., the “red dot” in Figure 1), instead of using the Dirichlet projection.  Empirically, this variant behaves very similarly to “SincAdaptNet + EA” (mean class-probability without GMM calibration), providing only marginal improvements. Without uncertainty modeling, misclassified trials tend to appear as outliers in the GMM clusters, which makes it difficult for the GMM alone to correct these errors based solely on the mean probability vectors. **We explicitly describe this new ablation and its implications in the revised ablation paragraph and in the updated Table 8**.
>
> **Point 1. Inclusion of Table A, discussion and future work.**
> **We have added the Table A to Appendix A (Related Work)** to provide readers with the necessary context regarding the performance of large-scale models vs. lightweight models in MI tasks. To prevent numbering conflicts with prior responses to other reviewers, we currently keep the label “Table A” in the appendix, and will renumber it in the camera-ready version if the paper is accepted.
>
> Following your suggestion, **we have also added a detailed discussion of why large scale pretraining has not yet brought clear benefits for handling MI variability.** Due to space constraints, we place it together with **Table A** in **Appendix A (Related Work)** to create a more coherent narrative. The main points are:
>
> - First, MI-induced EEG signals are inherently weak and highly variable compared to other paradigms like visual evoked potentials. Unlike P300 or SSVEP which show consistent patterns across subjects, MI relies on subjective imagination to trigger sensorimotor rhythm ERD/ERS. This results in subtle power changes in specific bands (e.g. $\mu$: 8–13 Hz, $\beta$: 18–30 Hz) that are often masked by noise. Moreover, these subtle power changes vary significantly across subjects. Some subjects demonstrate clear $\mu$-rhythm ERD/ERS while others show only minimal changes, which is the "BCI illiteracy" phenomenon [a]. This extreme weakness and variability make unified feature extraction challenging, and large-scale models may not capture these subtle subject-specific nuances without task-aligned pretraining.
>
> - Second, current pre-training strategies for EEG foundation models may face limitations. Most foundation models are trained with objectives like masked reconstruction or contrastive learning that are inherited from NLP and vision tasks, to learn general representations from unlabeled EEG data [b]. However, these objectives may not align with task-specific requirements. For instance, masked reconstruction encourages the model to recover missing signal segments based on contextual patterns. These patterns may emphasize global temporal structure that relates to noise rather than task-relevant features. In addition, many models mix diverse data without task/subject differentiation [c], leading to "averaged" representations of unclear utility. This explains why simple linear classifiers perform poorly on these pretrained representations in certain cases, where the features lack linearly separable task-relevant information [d]. Consequently, they require substantial non-linear adaptation during fine-tuning, which can negate the benefits of pretraining and risk overfitting on limited downstream datasets.
>
>  **In the Conclusion, we have also added the sentence**  “In future work, we commit to integrate BTTA-DG with large pretrained EEG models to explore whether the gains are additive.”
>
> **Point 2. Line 73 grammatical coherence.**
> Thank you for pointing out this grammatical issue. **We have corrected the phrasing in line 73 of the revised manuscript.**

---

> > ### Author Response · Authors · 2025-11-29
> > **Furthur Response [2/2] to Official Comment by Reviewer G2CU**
> >
> > (continued):
> >
> > Once again, we thank you for your invaluable guidance throughout this process. Your rigorous feedback and willingness to engage in discussion have been instrumental in elevating the quality of our paper. We hope that these clarifications and additions fully address your remaining questions and points. If any part of the response is unclear, we are happy to offer further clarification.
> >
> > ---
> > [a] Pauline Dreyer, et al. A large EEG database with users’ profile information for motor imagery brain-computer interface research. Scientific Data, 10(1):580, 2023.
> > [b] Yao Yuxuan, et al. Foundation models for EEG decoding: current progress and prospective research. Journal of Neural Engineering, 2025.
> > [c] Chi-Sheng Chen, et al. Large cognition model: Towards pretrained EEG foundation model. arXiv preprint arXiv:2502.17464, 2025.
> > [d] Anonymous, Are EEG Foundation Models Worth It? Comparative Evaluation with Traditional Decoders in Diverse BCI Tasks, Submitted to the 14th International Conference on Learning Representations, 2025, under review.

---

### Official Review · Reviewer_Av96 · 2025-10-31

**Soundness:** 3
**Presentation:** 3
**Contribution:** 3
**Rating:** 6
**Confidence:** 3

**Summary:**

The paper proposes BTTA-DG, a Bayesian Test-Time Adaptation (TTA) framework for motor imagery (MI) EEG decoding. The method aims to address cross-subject and cross-session variability by enabling gradient-free adaptation during inference. It consists of three core components:
(1) a Sinc-based adaptive network (SincAdaptNet) for frequency-selective representation learning,
(2) a Dirichlet feature projection that probabilistically models deep feature uncertainty, and
(3) a GMM-driven Bayesian inference mechanism that calibrates model predictions without updating weights.

The framework is theoretically motivated and computationally efficient. By introducing Dirichlet modeling into the EEG-TTA setting, the paper provides a probabilistic view of feature dynamics and uncertainty estimation. Experiments demonstrate competitive accuracy and improved interpretability compared with prior TTA approaches.

**Strengths:**

- The authors successfully introduce Dirichlet modeling into the EEG-TTA decoding paradigm and achieve strong results on four public datasets, validating the method’s empirical effectiveness.

- The proposed BTTA-DG framework leverages GMM-driven Bayesian inference to avoid parameter updates during test time, achieving fast inference speed and making it suitable for real-world BCI deployment.

- The model effectively mitigates cross-subject and cross-session variability, demonstrating that a single model can generalize to new subjects and sessions at test time without gradient updates.

**Weaknesses:**

- The paper employs SincAdaptNet for pretraining, which contains only four convolutional layers. Given the limited size of EEG datasets, it remains unclear how such a lightweight backbone can ensure strong representational capability. If the contribution mainly lies in “extracting task-specific frequency bands,” this may be insufficient as a novel methodological point. The authors are encouraged to clarify the specific purpose and benefits of frequency-band extraction in the context of TTA and practical BCI applications.

- The Dirichlet distribution is introduced to parameterize the probabilistic distribution over class predictions. However, since the scale parameter α₀ significantly affects the concentration of the distribution, the authors should explain how α₀ is chosen or tuned, and analyze its impact on model stability and uncertainty estimation.

- Although Table 7 reports the average inference time, it would be beneficial to include a comparison of runtime and memory usage, to quantitatively demonstrate the computational efficiency of the Dirichlet modeling and strengthen the claim of practical advantage.

- While the GMM-driven Bayesian inference is theoretically sound, the interaction between Dirichlet feature projection and GMM posterior fusion remains insufficiently explained. The authors should further clarify how the historical GMM distribution influences the Dirichlet-projected features during test time, and whether it could cause bias toward recent samples. Providing additional conceptual explanation or visualization would help improve clarity.

- The ablation study is not sufficiently comprehensive. A more systematic evaluation, such as testing under cross-session and cross-subject settings and examining whether the improvements are consistent across multiple datasets, would substantially strengthen the empirical validity and generalizability of the proposed framework.

**Questions:**

- The paper claims that BTTA-DG achieves lower computational overhead than existing EEG-TTA methods. Quantitative evidence—such as comparisons of inference time, FLOPs, or memory usage—would be helpful to substantiate this claim.

- The proposed online TTA framework appears to operate only with a batch size of 1. It would be useful to clarify whether this is a methodological constraint or a design choice, as well as the rationale for this setting and its consistency with real-world BCI scenarios where input configurations may vary.

- Further clarification is needed on how the “historical distribution” in the GMM-based Bayesian inference is estimated and updated during test time. If this distribution is continuously adapted, it may lead to potential distribution drift or overfitting toward recent samples.

-The paper presents visualizations of learnable Sinc filters and claims physiological interpretability. A quantitative evaluation of whether the learned frequency bands align with known EEG rhythms (e.g., μ, β, γ bands) would strengthen the interpretability argument.

---

> ### Author Response · Authors · 2025-11-22
> **Response [1/4] to Official Review by Reviewer Av96- Weakness 1**
>
> Dear Reviewer Av96,
>
> We thank you for your careful review and questions. We respond to each weakness and question below, and describe the additional analyses and experiments included in our revision.
>
> **Addressing Weaknesses**
>
> **1. Lightweight SincAdaptNet backbone and the role of frequency-band extraction:**
> We appreciate your feedback regarding the SincAdaptNet backbone and the role of frequency-band extraction. In order to avoid any potential challenges about the core novelty of this paper, we first respectfully clarify that the primary novelty of our work lies not only in the SincAdaptNet backbone, but also in the efficient gradient-free Bayesian test-time adaptation via Dirichlet feature projection and GMM-driven inference.
>
> **Representation capacity of the lightweight backbone**
> Although SincAdaptNet has only four convolutional layers, it is not a weak encoder in the MI-EEG setting. From a statistical perspective, MI decoding is largely governed by low dimensional structure in the time–frequency domain, for example μ and β rhythms over a small set of motor channels. Under limited data, very deep architectures tend to increase variance rather than capture additional task relevant structure. SincAdaptNet imposes a strong inductive bias by parameterizing each temporal filter with only two spectral parameters, low cut off and bandwidth, which encourages band limited solutions aligned with well known MI neurophysiology. This narrows the hypothesis space and yields a more favorable bias variance trade off. **We have added this clarification in Appendix B.**
>
> Empirically, SincAdaptNet outperforms the deeper EEG Conformer (Transformer-based) on 5 datasets across various settings, as shown in **Tables 2, 3, 4, 5, 10, 14, 16 and 17**, even when both models are used purely as source encoders. In addition, our survey on large scale pretrained EEG models shows that these models with millions of parameters do not consistently surpass lightweight architectures, despite their much larger capacity. We summarize this comparison in **Table A**.
>
> **Table A: Comparison of large-scale models vs lightweight models on BCIC-IV-2a-4class and BCIC-2b (LOSO).**
>
> | Scale                         | Methods     | Number of Parameters | BCIC-IV-2a-4class | BCIC-2b |
> |------------------------------|------------|----------------------|-------------------|---------|
> | Large-scale foundation models | BENDR [a]   | 4 M                  | 0.4899            | 0.7067  |
> |   | BIOT [b]    | 3.2 M                | 0.4590            | 0.6409  |
> |   | LaBraM [c]  | 5.8 M                | 0.5613            | 0.6851  |
> |   | EEGPT [d]   | 25 M                 | 0.5846            | 0.7212  |
> |   | CBraMod [e,f] | 4.0 M              | 0.5585            | 0.6735  |
> | Lightweight models           | EEGNet [g]  | 0.003 M              | 0.5685            | 0.7429  |
> |             | Conformer [h] | 0.55 M             | 0.5341            | 0.7361  |
> |             | SincAdaptNet | 0.0015 M           | 0.5764            | 0.7632  |
>
> **Novelty of Sinc-based frequency filters**
> On the methodological side, we agree that simply “extracting task-specific frequency bands” would not by itself be sufficiently novel. However, our Sinc Conv layer goes beyond standard temporal convolutions that only implicitly encode spectral preferences through  manually tuned kernel lengths. Instead, it learns only the low cut and bandwidth parameters of each band pass filter and analytically constructs the time domain kernel. This makes every filter directly interpretable in the frequency domain. As illustrated in **Figures 2 and 6–13**, the learned passbands cluster around the canonical μ (8–13 Hz), β (13–30 Hz) and low γ (30–45 Hz) ranges. **In the revision we further provide a quantitative analysis in Table 6 and Appendix L, showing that more than 92% of the learned filters lie within a small tolerance of these bands, which supports the strong physiological plausibility of the learned filters.**
>
> **Benefits of frequency-band extraction for TTA and practical BCIs**
> Finally, we clarify why frequency band extraction is particularly beneficial for TTA and for practical BCIs. The learnable Sinc filters isolate the physiologically critical μ and β bands and related low γ components, while filtering out much of the broadband and muscle related noise present in raw EEG. This yields a more stable and less ambiguous sequence of predictive embeddings over each trial. Feeding these cleaner embeddings into the accelerated fixed point Dirichlet estimation leads to more reliable Dirichlet parameters, which are exactly the features that BTTA DG adapts on. In other words, band limited features directly improve the quality of the probabilistic representation that drives our TTA module, enhancing robustness to non-stationarity in realistic online BCI scenarios while keeping the overall system efficient and interpretable. **We have also added this clarification in Appendix B.**

---

> ### Author Response · Authors · 2025-11-22
> **Response [2/4] to Official Review by Reviewer Av96- Weaknesses 2 & 3**
>
> **Addressing Weaknesses (continued)**
>
> **2. Selection of Dirichlet scale parameter α₀:**
> Thank you for raising this important point. We apologize for not explaining the treatment of $\alpha_0$ more clearly in the original submission. In our framework, $\alpha_0$ is never selected or tuned by hand. The Dirichlet parameters $\alpha=\left(\alpha_1,\ldots,\alpha_{\left|\mathcal{L}\right|}\right)$ (and thus $\alpha_0=\sum_{i}\alpha_i$) are obtained purely by maximum likelihood estimation from the sequential predictive embeddings $X$ of each trial, using an accelerated fixed-point algorithm (**see Appendix C**). In other words, $\alpha_0$ is a data-driven sufficient statistic, not a free hyperparameter.
>
> Conceptually, $\alpha_0$ directly encodes the predictive uncertainty rather than arbitrarily influencing it. A large $\alpha_0$ corresponds to highly consistent per-time-step predictions (low variance in the temporal categorical embeddings), yielding a concentrated Dirichlet and high confidence. Conversely, a small $\alpha_0$ arises when the predictions fluctuate strongly over time, yielding a diffuse Dirichlet and high uncertainty. This trial-specific, MLE-based estimate of $\alpha_0$ is precisely what makes the Dirichlet representation stable and informative for TTA, without introducing extra hyperparameters. **In the revised manuscript, we explicitly state in Section 2.2 that $\alpha_0$ is estimated via a fixed-point algorithm (not tuned) and clarify its role as an uncertainty indicator reflecting the temporal consistency of the model’s predictions.**
>
> **3. Runtime and memory usage:**
> Thank you for pointing out the need to more explicitly quantify runtime and memory usage. First, **Table 7** already reports the average inference time per trial, which directly corresponds to runtime. As shown there, BTTA-DG requires 15.7 ms per trial, which is lower than recent EEG-TTA baselines. This indicates that adding Dirichlet projection and GMM based Bayesian inference does not introduce substantial latency.
>
> Second, in response to your comment on memory usage, **we now provide an explicit analysis in Appendix O (Computational Complexity).** The adaptation trials is stored as low dimensional Dirichlet parameters in a memory bank with space complexity $O\left(M\cdot d\right)$, where $M$ is the bank size and $d=\left|\mathcal{L}\right|$ is the number of classes. Concretely, for a two class MI task with $d=2$ and two memory buffers of $M=1000$ trials per subject, the cost is
> $2 \text{ buffers}\times1000\mathrm{\ trials}\times2\mathrm{\ floats/trial}\times4\mathrm{\ bytes/float\ 32}=16\ \text{KB}$ per subject. With the same $M$, the per subject memory remains on the order of 64 KB, and the GMM parameters add only a few kilobytes.
>
> Moreover, to address your request for a more quantitative runtime comparison, **we augmented Table 7 to report estimated FLOPs per trial alongside inference time.** As summarized in the revised **Table 7**, BTTA-DG uses 141.6 M FLOPs per trial, while gradient based methods such as Tent, CoTTA, SAR and OTTA require substantially more operations. Combined with the memory analysis above, these additions provide a concrete and quantitative demonstration that the BTTA-DG method remains computationally lightweight and practically deployable.

---

> ### Author Response · Authors · 2025-11-22
> **Response [3/4] to Official Review by Reviewer Av96- Weaknesses 4 & 5**
>
> **Addressing Weaknesses (continued)**
>
> **4. Interaction between Dirichlet feature projection and GMM posterior fusion; bias to recent samples**
> First, the Dirichlet feature projection and the GMM operate at different stages. For each test trial, we obtain the sequential embeddings $X$ from SincAdaptNet. The Dirichlet parameters $\alpha=\ (\alpha_1,\ldots,\alpha_{\left|\mathcal{L}\right|})$ are then estimated solely from this trial via maximum-likelihood (accelerated fixed-point iterations). This yields a trial-specific Dirichlet feature. Crucially, the GMM does not modify this Dirichlet feature. The historical GMM is then fitted on the collection of past Dirichlet feature stored in the memory bank to model class-specific likelihood $p_{\mathrm{GMM}}\left(\alpha\mid y\right)=\sum_{k=1}^{K}\pi_{y,k}  \mathcal{N}\left(\alpha;\mu_{y,k},\Sigma_{y,k}\right).$  At test time, the calibrated prediction combines the network’s output and likelihood $p_{\mathrm{cal}}(y∣\alpha_{\text{MLE}})∝p_\theta(y)\  p_{\mathrm{GMM}}(\alpha_{\text{MLE}}∣y),$  so the historical GMM influences only the posterior over labels, not the Dirichlet-projected feature itself. **This interaction is now explicitly described in Section 2.2, subsection “2) GMM-driven Bayesian Inference for Gradient-Free Calibration”.**
>
> Second, regarding bias toward recent samples, the memory bank and its update rules are designed for controlled adaptivity. We use a fixed-size buffer, discard the oldest entries when full, and only insert trials that satisfy both a confidence threshold $\tau_{\mathrm{conf}}$ and an entropy threshold $\tau_{\mathrm{ent}}$. Noisy or highly uncertain recent trials are therefore rejected and do not affect the GMM, while very old trials are gradually forgotten. Each inserted trial contributes at most \(1/M\) to the GMM fitting, so a small set of recent trials cannot dominate the GMM posterior fusion. **This rationale is now clearly stated in Section 2.2, subsection 2), in order to address your concern about the bias control in recent samples.**
>
> **5. Ablation study across settings and datasets:**
> Thank you for highlighting the need for a more systematic ablation, especially across different settings and datasets. **In the revised manuscript, we have substantially extended the ablation study in updated _Table 8_ to address this concern.**
> Concretely, we now evaluate five variants on BNCI2014001 under both cross-session and cross-subject settings, and on BNCI2014002, BNCI2015001, and SHU MI under cross-subject settings: (i) SincAdaptNet (Source Only), (ii) BTTA-DG without EA (Dirichlet + GMM only), (iii) SincAdaptNet + EA, (iv) SincAdaptNet + EA + Dirichlet, and (v) full BTTA-DG (EA + Dirichlet + GMM).
>
> As summarized in the revised **Table 8**, EA consistently improves over the source-only model in all settings, and BTTA-DG w/o EA also yields clear gains, showing that the Dirichlet+GMM module has standalone adaptation capability when EA cannot be applied. Adding Dirichlet projection on top of EA further improves performance without any test-time adaptation, while the full BTTA-DG achieves the best accuracy on every dataset and setting, with gains of about 2–6% over the SincAdaptNet source baseline. These results demonstrate that the benefits of our framework are robust across both cross-session and cross-subject regimes and generalize well across multiple MI datasets.
>
> ---
> [a] Demetres Kostas, et al. BENDR: Using transformers and a contrastive self-supervised learning task to learn from massive amounts of eeg data. Frontiers in Human Neuroscience, 15:653659, 2021.
> [b] Chaoqi Yang, et al. BIOT: Biosignal transformer for cross-data learning in the wild. Advances in Neural Information Processing Systems, 36:78240–78260, 2023.
> [c] Wei-Bang Jiang, et al. Large brain model for learning generic representations with tremendous eeg data in bci. In The Twelfth International Conference on Learning Representations, 2024.
> [d] Guangyu Wang, et al. EEGPT: Pretrained transformer for universal and reliable representation of eeg signals. Advances in Neural Information Processing Systems, 37:39249–39280, 2024.
> [e] Jiquan Wang, et al. CBramod: A criss-cross brain foundation model for EEG decoding. In The Thirteenth International Conference on Learning Representations, 2025.
> [f] Ang Li, et al. Comet: A contrastive-masked brain foundation model for universal EEG representation. arXiv preprint arXiv:2509.00314, 2025.
> [g] Vernon J Lawhern, et al. EEGNet: a compact convolutional neural network for EEG-based brain–computer interfaces. Journal of neural engineering, 15(5):056013, 2018.
> [h] Song, Y., et al. EEG conformer: Convolutional transformer for EEG decoding and visualization. IEEE Transactions on Neural Systems and Rehabilitation Engineering, 31, 710-719.

---

> ### Author Response · Authors · 2025-11-22
> **Response [4/4] to Official Review by Reviewer Av96- Questions**
>
> **Addressing Questions**
>
> **1. Inference time, FLOPs, and memory usage:**
> Thank you for raising this point. Please see our response to Weakness 3.
>
> **2. Batch Size of 1:**
> Our framework is not restricted to a batch size of 1. Our online MI-TTA setting assumes sequential single-trial arrival (batch size = 1), consistent with prior EEG-TTA works such as T-TIME [i]. Compared with mini-batch settings, this single-trial setting is both more realistic and more challenging, because many TTA methods (e.g., batchnorm–based approaches like BN-adapt) rely on batch statistics and degrade when only one sample is available. In contrast, our method works robustly with batch size of 1 thanks to the use of layer normalization and a probabilistic GMM over Dirichlet features instead of batch statistics. The same adaptation mechanism can be directly applied to larger mini-batches without any modification. **We have added this explanation in Appendix E (Implementation Details).**
>
> **3. Historical distribution update in the GMM-based Bayesian inference:**
> Thank you for raising this important point. In our framework, the “historical distribution” is modeled as a class-specific GMM fitted over Dirichlet parameters stored in a fixed-size memory bank. For each class, this bank keeps at most \(M\) past trials. When it is full, the oldest entry is discarded as a new one is added, yielding a dynamic estimate of the target distribution. A new trial is inserted into the bank only if its Dirichlet posterior is both confident and low-entropy, so noisy or ambiguous samples are explicitly excluded. After insertion, we refit the GMM on the entire bank using standard Expectation-Maximization (EM). This is feasible because the Dirichlet parameters are very low-dimensional, so EM refitting has small overhead, whereas sophisticated incremental or online clustering methods tend to be more complex and empirically less stable for our setting.
>
> Regarding drift and overfitting to recent trials, the memory bank and update rules are designed to achieve controlled adaptivity. We maintain a fixed-size buffer, discarding the oldest entries when full, and we only insert trials whose Dirichlet posteriors satisfy both a confidence threshold $\tau_{\mathrm{conf}}$ and an entropy threshold $\tau_{\mathrm{ent}}$. Thus, highly uncertain or noisy recent trials are rejected and do not affect the GMM, while very old trials are gradually removed so that the model does not remain anchored to an outdated distribution. Each accepted trial contributes at most \(1/M\) to the GMM fitting, so a small subset of recent samples cannot dominate the posterior fusion. **These clarifications are now described more explicitly in the revised Section 2.2.**
>
> **4. Quantitative assessment of the learned frequency bands:**
> Thank you for this insightful suggestion. In the original version we provided a qualitative analysis of the learned Sinc filters in **Figure 2** and in **Appendix K**, where we plotted the frequency responses for each subject and showed that most filters concentrate around the μ, β and low-γ ranges.
>
> **In the revised version we now add a quantitative evaluation in Table 6 and Appendix L.** Concretely, for each learned Sinc filter we compute its lower and upper cut-off frequencies and check whether its passband lies within a ±2 Hz tolerance around canonical motor-imagery rhythms, i.e., μ (8–13 Hz) [j], β (13–30 Hz) [k], and low-γ (30–45 Hz) [l]. **Table 6** reports the proportion of filters per subject that fall into each band. On average, about 56% of filters lie in the μ/β range, about 36% in the low-γ band, and fewer than 8% outside these MI rhythms. This quantitative evidence supports our interpretability claim that SincAdaptNet learns physiologically meaningful sensorimotor frequency bands.
>
> ---
>
> We thank you again for the time and effort you have devoted to reviewing our work. Your insightful comments have significantly enhanced the quality of the paper. We hope that the additional analyses and experiments have addressed your concerns. We would be most appreciative if you could consider reflecting this in your final rating. If any issues remain unclear, we would be delighted to provide further clarification.
>
>
> [i] Siyang Li, et al. T-TIME: Test-time information maximization ensemble for plug-and-play BCIs. IEEE Transactions on Biomedical Engineering, 71 (2):423–432, 2023.
> [j] Dennis J McFarland, et al. Mu and beta rhythm topographies during motor imagery and actual movements. Brain topography, 12:177–186, 2000.
> [k] Gert Pfurtscheller, et al. Mu rhythm (de) synchronization and EEG single-trial classification of different motor imagery tasks. NeuroImage, 31(1):153–159, 2006.
> [l] Álvaro Sabater-Gárriz, et al. Affective touch enhances low gamma activity during hand proprioceptive perception in children with different neurodevelopmental conditions. Frontiers in Human Neuroscience, 19:1538428, 2025.

---

### Official Review · Reviewer_4sdH · 2025-11-01

**Soundness:** 3
**Presentation:** 3
**Contribution:** 3
**Rating:** 6
**Confidence:** 3

**Summary:**

This paper presents a Bayesian test-time adaptation framework for EEG motor imagery decoding that addresses cross-subject and cross-session generalisation challenges. It leverages Euclidean Alignment (EA) to align EEG trials into a common domain, followed by a gradient-free Bayesian adaptation on low-dimensional Dirichlet-distributed sequential embeddings, extracted from pretrained SincAdaptNet models.  This method offers a gradient-free approach that avoids catastrophic forgetting and adapts robustly during test time.

**Strengths:**

The method emprically demonstrates its consistent outperformance across four well-known and diverse motor imagery EEG datasets with rigorous leave-one-subject-out and cross-session validation.

The projection of high-dimensional sequential EEG embeddings onto a compact Dirichlet-distributed parameter space is an innovative approach to capture predictive uncertainty and temporal evidence concentration.

The learned filters and Dirichlet feature spaces align with neurophysiological expectations, which supports the model’s interpretability and trustworthiness

**Weaknesses:**

The lack of an ablation study without EA precludes understanding of the standalone adaptation capabilities. This reliance could limit practical application scenarios where EA cannot be applied, e.g., variable or missing channels, high artifact environments.


The current approach processes entire trials for adaptation and inference, which is less practical for online BCIs requiring low latency and continuous feedback.

**Questions:**

Given the practical constraints on EEG hardware, benchmarking performance on datasets with fewer EEG channels is advisable. While BNCI and SHU datasets are well established, inclusion of standard datasets like BCI Competition IV Dataset 2b (3 channels) or datasets with low-density configurations would strengthen practical relevance.

For more realistic online BCI applications, investigating windowing or sliding window strategies, which segment data into shorter temporal windows for quick iterative adaptation and inference, would address practical BCI applications.

A recommended discussion point to include in the paper is addressing scalability and memory cost concerns for large-scale deployment across thousands of individuals, given the fact that each trial from each participant is represented and stored.

---

> ### Author Response · Authors · 2025-11-22
> **Response [1/2] to Official Review by Reviewer 4sdH- Weaknesses**
>
> Dear Reviewer 4sdH,
>
> We thank you for your detailed review and valuable suggestions. Below, we respond to your points one by one and describe the additional analyses and experiments we have incorporated into the revised manuscript:
>
> **Addressing Weaknesses**
>
> **1. Lack of ablation without EA:**
> We appreciate this point, as it highlights the importance of the BTTA-DG for real-world robustness. Before applying the adaptive module, each EEG trial is aligned to a common domain using Euclidean Alignment (EA), which is a normal preprocessing step in EEG-TTA studies [a][b]. EA allows our Dirichlet–GMM module to focus on shifts in the distribution of time-varying categorical embeddings. In real-world scenarios with missing channels or strong artifacts, EA can still be made applicable by standard preprocessing, for example imputing missing channels via interpolation or removing artifact components via ICA.
>
> In addition, we fully agree that it is important to understand the standalone adaptation capability of BTTA-DG when EA cannot be used. **In the revised manuscript, we therefore added a new “BTTA-DG w/o EA” condition to the ablation study and expanded it across BNCI2014001 (cross-session and cross-subject), BNCI2014002, BNCI2015001 and SHU MI (**Table 8**).** This variant removes EA but keeps the Dirichlet projection and GMM-based inference. The results show that BTTA-DG w/o EA consistently improves over SincAdaptNet (Source Only) by roughly 1–2% on all datasets, confirming that the Dirichlet+GMM module has genuine standalone adaptation ability. EA further boosts performance, and the full BTTA-DG remains the best model in every setting. **We explicitly describe these findings in the updated ablation paragraph and refer the reader to the new **Table 8** in the revised PDF.**
>
> **2. Whole-trial adaptation vs. sliding-window online BCIs:**
> Our method is well-suited for low-latency online use. In our experiments each trial lasts 4–5 seconds, while the total computational overhead of BTTA-DG (forward pass + Dirichlet MLE + GMM inference) is only 15.7 ms per trial (**Table 7**), which is below one percent of the trial duration. Moreover, our online TTA setting operates with batch size equal to 1, where trials arrive sequentially and are adapted individually. This protocol is procedurally equivalent to a sliding-window paradigm with a 4–5 second window and confirms that the framework is not inherently tied to offline, multi-trial batching.
>
> To further address your concern, **we have added a new section “Sliding-window online adaptation” in **Appendix I** with **Table 16**.** There we evaluate BTTA-DG on BNCI2014001 in the cross-subject setting but segment each 4-second trial into non-overlapping 1-second windows and perform adaptation and inference per window. As reported in **Appendix I Table 16**, all methods show lower accuracy than in the full-trial case. Importantly, BTTA-DG still attains the best average performance (76.49%) and improves over the SincAdaptNet source model (74.52%) with statistically significant gains on several subjects and on the average. This indicates that our Dirichlet–GMM adaptation mechanism remains effective on short windows and that BTTA-DG can operate in low-latency, continuous-feedback BCIs rather than being restricted to whole-trial processing.
>
> ---
> [a] Siyang Li, Ziwei Wang, Hanbin Luo, Lieyun Ding, and Dongrui Wu. T-TIME: Test-time information maximization ensemble for plug-and-play BCIs. IEEE Transactions on Biomedical Engineering, 71 (2):423–432, 2023.
> [b] Wimpff, M., et al. (2024). Calibration-free online test-time adaptation for electroencephalography motor imagery decoding. 2024 12th International Winter Conference on Brain-Computer Interface (BCI), 1–6.

---

> ### Author Response · Authors · 2025-11-22
> **Response [2/2] to Official Review by Reviewer 4sdH- Questions**
>
> **Addressing Questions:**
>
> **1. Performance with low-Channel datasets:**
> Thank you for this suggestion, which indeed strengthens the practical relevance of our work. We would first like to note that the main experiments cover relatively low-density configurations, using datasets with 22 channels (BNCI2014001), 15 channels (BNCI2014002), 13 channels (BNCI2015001) and 32 channels (SHU MI).  To explicitly address very low channel counts, **we have added a new experiment on BCI Competition IV 2b, which has only three EEG channels.** We adopt the same LOSO cross-subject protocol and keep all architecture, training and adaptation hyperparameters unchanged except for the input channel dimension. **The new results are reported in **Appendix G, Table 14**.** As shown there, BTTA-DG attains the highest average accuracy of 78.76% and yields statistically significant gains over the SincAdaptNet source model, despite the extremely low electrode density. This confirms that the proposed adaptation mechanism remains effective under minimal channel configurations and supports the robustness and practical suitability of BTTA-DG for low-channel BCI hardware.
>
> **2. Sliding-window adaptations for online BCIs:**
> This point is closely related to our response to Weakness 2. In brief, BTTA-DG already supports online use because the total per trial overhead is only 15.7 ms, which is negligible compared with 4–5 second trials, and our online TTA operates with batch size 1 so trials or windows are adapted sequentially in the same way as a sliding window BCI. **To make this explicit we added a sliding window experiment in **Appendix I** with **Table 16**,** where 4 second trials are segmented into 1 second windows and adapted one by one, confirming that the framework is compatible with realistic window based online BCIs.
>
> **3. Scalability and memory cost for large-scale deployment:**
> Thank you for raising this practical point. In our intended use case, the backbone is trained once on many source subjects, and BTTA-DG is then run per target user (or a small number of target users). The adaptation state is therefore kept only for the active target users, not for thousands of subjects at the same time, which limits the overall memory usage.
>
> **In the revised paper we now quantify memory usage explicitly in **Appendix O** (Computational Complexity).** The memory bank stores only low-dimensional Dirichlet parameters with space complexity $O(M \cdot d)$, where $M$ is the bank size and $d = |\mathcal{L}|$ is the number of classes. For a 2-class task ($d=2$) with 2 memory buffers of $M=1000$ trials per subject, the cost is $2 \text{ buffers} \times 1000 \text{ trials} \times 2 \text{ floats/trial} \times 4 \text{ bytes/float32} = 16 \text{ KB}$ per subject. Even for a 4-class task, the total per-subject memory is on the order of 64 KB, with GMM parameters adding only a few kilobytes. Since this is stored for adapted users, the method remains lightweight even when deployed at scale.
>
> To complement the memory analysis, **we also added FLOPs estimates per trial on BNCI2014001 in the updated **Table 7**, alongside the inference time.** BTTA-DG requires 141.6M FLOPs and 15.7 ms per trial, while gradient-based TTA methods have more FLOPs and higher latency. These numbers indicate that BTTA-DG is both memory efficient and computationally affordable for large-scale deployment.
>
> ---
>
> We hope these responses and new experiments have fully addressed your concerns. We are grateful for the opportunity to improve our paper based on your feedback. If our response has resolved your questions, we would be most grateful if you could consider improving rating, as this is of vital importance to us. If you have any further questions, please do not hesitate to ask; we remain at your disposal at all times.

---

### Author Response · Authors · 2025-11-24
**Summary of our response**

We sincerely thank all reviewers for their exceptionally thorough and constructive feedback. We have made extensive revisions addressing all concerns. Below we summarize the main changes, following the logical flow of the revised manuscript:

**SincAdaptNet Architecture Justification:**
- (Reviewers Av96, G2CU) Added comprehensive motivation for SincAdaptNet including inductive bias analysis and TTA benefits (**Appendix B**)
- (Reviewers Av96, G2CU) Provided literature comparison with large-scale pretrained EEG models (BENDR, BIOT _NeurIPS'23_, LaBraM _ICLR'24_, EEGPT _NeurIPS'24_, CBraMod _ICLR'25_) (**Table A** in rebuttal)
- (Reviewer G2CU) Updated Related Work to discuss supervised fine-tuning approaches (**Appendix A**)

**Methodological Clarifications:**
- (Reviewers Av96, G2CU) Replaced "deep features" with "temporal predictive embeddings" throughout to accurately reflect our approach (**Section 2.2**)
- (Reviewer Av96) Clarified that $ \alpha_0 $ is MLE-estimated via accelerated fixed-point algorithm, not a tuned hyperparameter (**Section 2.2**)
- (Reviewer Av96) Explained Dirichlet-GMM interaction mechanism and bias control for recent samples (**Section 2.2**)
- (Reviewer Av96) Clarified batch size = 1 as realistic online TTA setting, not a methodological constraint (**Appendix E**)
- (Reviewer Av96) Detailed historical distribution update mechanism in GMM-based Bayesian inference (**Section 2.2**)

**New Experiments:**
- (Reviewer 4sdH) Added low-channel adaptation on BCI Competition IV 2b (3 channels) (**Appendix G, Table 14**)
- (Reviewer 4sdH) Added sliding-window online adaptation with 1-second windows (**Appendix I, Table 16**)
- (Reviewer G2CU) Added within-subject 3-fold cross-validation to analyze intrinsic subject decodeability (**Appendix J, Table 17**)

**Interpretability Enhancement:**
- (Reviewers Av96, G2CU) Added quantitative filter alignment analysis showing 92%+ of learned filters align with μ/β/γ MI rhythms (**Table 6, Appendix L**)

**Expanded Computational Efficiency Analysis:**
- (Reviewers 4sdH, Av96) Expanded **Table 7** with per-trial FLOPs alongside inference time
- (Reviewers 4sdH, Av96) Added explicit memory usage analysis (**Appendix O**)
- (Reviewer G2CU) Explained why BN-adapt is fastest (only recomputes batch-norm statistics)

**Expanded Ablation Study:**
- (Reviewers 4sdH, Av96, G2CU) Added "BTTA-DG w/o EA" variant and extended the ablation study of five variants across settings and datasets (**Table 8**)

**Presentation Improvements:**
- (Reviewer G2CU) Fixed conclusion paragraph grammar and clarified empirically-supported claims
- (Reviewer G2CU) Unified Figure 4 y-axes for better hyperparameter sensitivity comparison
- (Reviewer G2CU) Reordered tables/figures to match text flow
- (Reviewer G2CU) Moved class imbalance analysis to main Results section (**Table 9**)
- (Reviewer G2CU) Corrected Figure 1 caption

We are very grateful for the time and care all reviewers invested in this process. Your comments have led to substantial improvements in clarity, empirical support, and presentation. We hope these revisions address your concerns and would be sincerely thankful if you could take them into account in your final ratings. If any aspect of our response or revision remains unclear, we would be glad to provide further clarification at any time.

---

> ### Author Response · Authors · 2025-11-29
> **Further Summary of Rebuttals and Discussions**
>
> Dear Reviewers, AC, SAC, and PC,
>
> We deeply appreciate your continued engagement and hard work during this review process, particularly given the recent challenges posed by platform issue. In addition to the “Summary of our response” above, we would like to briefly summarize how the main concerns of each reviewer were addressed during the rebuttal and discussion phase, so that you have a succinct, post-rebuttal view when forming your final assessment.
>
> **Response to Reviewer 4sdH:**
> We addressed all major concerns including: (1) added "BTTA-DG w/o EA" variant demonstrating standalone adaptation capability across five settings (**Table 8**); (2) conducted new low-channel experiment on BCI Competition IV 2b (3 channels) showing maintained effectiveness (**Appendix G, Table 14**); (3) performed sliding-window adaptation experiment confirming real-time BCI compatibility (**Appendix I, Table 16**); (4) provided comprehensive computational analysis with FLOPs, inference time, and memory usage (**Table 7, Appendix O**). Unfortunately, this reviewer did not have the opportunity to respond to our rebuttal.
>
> **Response to Reviewer Av96:**
> We comprehensively addressed all concerns regarding: (1) SincAdaptNet's representation capacity with statistical justification and empirical comparisons (**Appendix B, Table A**); (2) clarified that $\alpha_0$ is MLE-estimated, not tuned (**Section 2.2**); (3) detailed Dirichlet-GMM interaction and bias control mechanisms (**Section 2.2**); (4) explained historical distribution update in GMM-based Bayesian inference (**Section 2.2**); (5) clarified batch size = 1 as realistic online TTA setting instead of methodological constraint (**Appendix E**); (6) expanded ablation study across multiple datasets and settings (**Table 8**); (7) added quantitative filter alignment analysis showing 92%+ alignment with MI rhythms (**Table 6, Appendix L**); (8) expanded computational efficiency analysis with FLOPs, inference time, and memory usage (**Table 7, Appendix O**). This reviewer also did not have the opportunity to comment on our response.
>
> **Response to Reviewer G2CU:**
> We are deeply grateful for this reviewer's continued engagement. In our initial rebuttal, we addressed major concerns by: (1) replacing "deep features" with "temporal predictive embeddings" to accurately reflect our approach; (2) justifying SincAdaptNet backbone choice with comprehensive analysis (inductive bias, TTA benefits in **Appendix B** and comparison with large-scale pretrained EEG models in **Table A**); (3) adding within-subject cross-validation (**Appendix J, Table 17**); (4) expanding ablation study (**Table 8**); (5) improving presentation throughout.
> The reviewer responded positively, acknowledging that the "deep feature" issue was "a phrasing problem" and stating that our contribution is "significant for real-world BCIs, as well as methodologically novel and experimentally well-supported." They raised their score from 4 to 6 and expressed openness to further increase, noting "most of my other concerns have also been either remedied in the new version, or clarified in your response."
> During discussion phase, we further addressed their remaining quesions and points by: (1) adding the requested "SincAdaptNet + EA + GMM" ablation variant to **Table 8**, which performs GMM calibration directly on mean class-probabilities without Dirichlet projection; (2) including **Table A** (comparison with large-scale pretrained models) in **Appendix A** with detailed discussion of why large-scale pretraining has not yet benefited MI tasks; (3) adding the suggested future work direction in the Conclusion; (4) correcting the grammatical error in line 73. Note that we maintain the label "Table A" in the current Appendix A to prevent numbering conflicts with prior responses to other reviewers, and will renumber it appropriately in the camera-ready version if accepted.
> We are sincerely grateful to Reviewer G2CU for their sustained and constructive engagement throughout this process, which has been instrumental in substantially strengthening our manuscript.
>
> **Respectful Request to ACs and Reviewers:**
> Given the premature termination of the discussion period, we would be deeply grateful if the ACs could consider how the reviewers' impressions would likely have changed had they been able to review our comprehensive responses and extensive revisions.
>
> We thank everyone for their invaluable contributions to improving this work. Your rigorous feedback has resulted in a substantially stronger manuscript with clearer methodology, better-supported claims, more comprehensive experiments, and improved presentation.
>
> Sincerely,
>
> The Authors

---

### Meta-Review · Area_Chair_AG9A · 2026-01-08

**Summary:**

The paper proposes a test-time adaptation method for cross-subject and cross-session motor imagery decoding, which is efficient and gradient-free.

**Strengths**

1. The idea of projecting EEG embeddings to Dirichlet parameters is interesting.

2. The performance is consistently improved.

3. The method achieves fast inference.

4. Neurophysiological evidence of the learned filters and feature spaces is provided.

**Weaknesses**

1. SincAdaptNet is rather shallow, which needs justification regarding the model choice and its good performance.

2. Benefits of frequency-band extraction and novelty beyond that need to be clearer.

3. More ablation studies are needed.

**Reviewer Concerns:**

1. Explanation and additional results seem to resolve the issue.

2. Additional results showing plausibility of learned filters are provided, which would resolve the issue at least partly.

3. Several ablation results including specific experiments mentioned by the reviewers are added.

**Reviewer Scores:**

Reviewer 4sdH would remain positive.

Reviewer Av96 would remain positive.

Reviewer G2CU replied with increasing the score to 6.

---

### Decision · Program_Chairs · 2026-01-26

Accept (Poster)